# MIXTURE OF LORA EXPERTS

**Xun Wu**[1,2]*, **Shaohan Huang**[1,✉], **Furu Wei**[1]
[1]Microsoft Research Asia   [2]Tsinghua Univeristy
wuxun21@mails.tsinghua.edu.cn; {shaohanh, fuwei}@microsoft.com

## ABSTRACT

Low-Rank Adaptation (LoRA) (Hu et al., 2021) has emerged as a pivotal technique for fine-tuning large pre-trained models, renowned for its efficacy across a wide array of tasks. The modular architecture of LoRA has catalyzed further research into the synergistic composition of multiple trained LoRAs, aiming to amplify performance across various tasks. However, the effective composition of these trained LoRAs presents a formidable challenge: (1) Linear arithmetic composition can lead to the diminution of the generative capabilities inherent in the original pre-trained models or the distinctive attributes of the individually trained LoRAs, potentially resulting in suboptimal outcomes. (2) Reference tuning-based composition exhibits limitations in adaptability and incurs significant computational costs due to the requirements to retrain a large model. In response to these challenges, we propose **Mixture of LoRA Experts** (**MoLE**). MoLE treats each layer of trained LoRAs as a distinct expert and implements hierarchical weight control by integrating a learnable gating function within each layer to learn optimal composition weights tailored specifically to the objectives of a given domain. MoLE not only demonstrates enhanced performance in LoRA composition but also preserves the essential flexibility necessary for effective composition of trained LoRAs with minimal computational overhead. Extensive experiments conducted in both Natural Language Processing (NLP) and Vision & Language (V&L) domains validate the effects of MoLE. Our code are available at https://github.com/yushuiwx/MoLE.git.

## 1   INTRODUCTION

Recent advances in deep learning have been driven by large-scale pre-trained models such as OPT (Zhang et al., 2022), LLaMA (Touvron et al., 2023) in the Natural Language Processing (NLP) domain and CLIP (Radford et al., 2021a), DALL·E 2 (Ramesh et al., 2022) in the Vision & Language (V&L) domain. These models show outstanding performance across various tasks when fine-tuned on down-stream datasets, but their increasing size entails significant computational costs for full fine-tuning. To mitigate this, LoRA (Hu et al., 2021) is introduced. By freezing the pretrained model weights and injecting trainable rank decomposition matrices, LoRA is proven to be an effective fine-tuning methodology in scenarios with constrained computational resources (Lester et al., 2021; An et al., 2022).

While LoRA serves as plug-and-play plugins for pretrained models, recent initiatives explores the composition of separate trained LoRAs to achieve joint generation of learned characteristics (Huang et al., 2023; Zhang et al., 2023; Ruiz et al., 2023). However, these efforts may encounter several challenges. As shown in Figure 2 (a), linear arithmetic composition (Zhang et al., 2023; Huang et al., 2023; Han et al., 2023) composes trained LoRAs

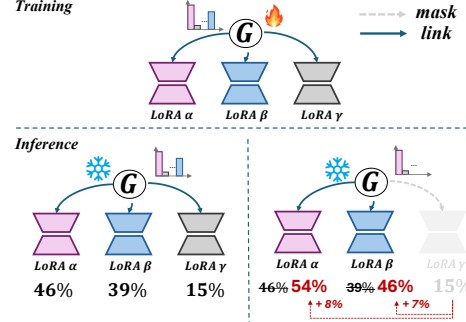

Figure 1: **Workflow of MoLE**. In the training phase, MoLE predicts weights for multiple LoRAs. In the inference phase, MoLE can allocate weights to multiple LoRAs, or, without altering the gating weights, achieve a more flexible LoRA composition by masking out undesired LoRAs and recalculating and distributing weights proportionally.

---

*Contribution during internship at Microsoft. ✉ Corresponding Author.

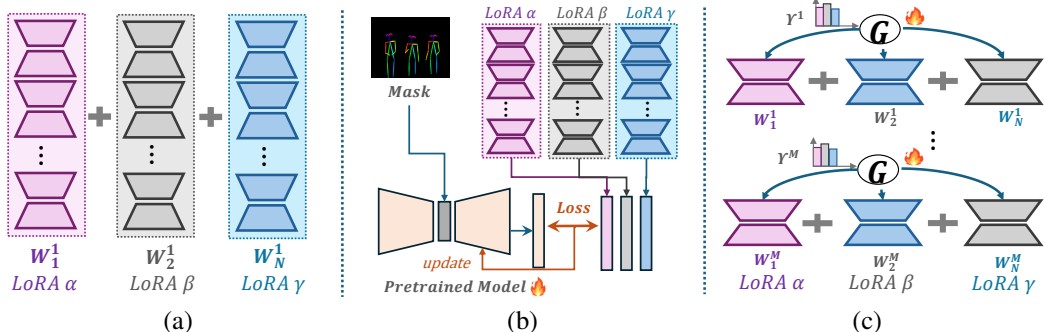

(a)            (b)            (c)

Figure 2: Overview of LoRA composition methods: (a) Linear arithmetic composition (Eq.2), which commonly applies the same composition weight $W_i$ to all layers of the $i^{th}$ LoRA. (b) Reference tuning-based composition involves retraining a large model by integrating outputs from multiple LoRAs using manually-crafted mask information. (c) Our MOLE, which learns a distribution $\Upsilon^j$ for the $j^{th}$ layer of LoRAs to determine the composition weight $W_i^j$.

directly. However, composing multiple LoRAs (typically $\geq 3$) can impair the generative performance of pre-trained models. To mitigate this, weight normalization is applied prior to the composition, but may erase the unique characteristics of individual trained LoRAs as the composing weight of each LoRA is reduced (refer to Observation 1 in § 3.1). Another approach, as depicted in Figure 2 (b), known as reference tuning-based composition (Gu et al., 2023), is tailored for the V&L domain and achieves superior performance. However, it is limited in terms of LoRA flexibility due to the utilization of manually-designed masks and involves substantial training costs, necessitating a full model retraining. In light of the above situation, an important question arises:

> *How can multiple trained LoRAs be composed dynamically and efficiently, while preserving all their individual characteristics?*

To address that issues, we introduce **M**ixture **o**f **L**oRA **E**xperts (**MOLE**). Recognizing that individual layers of a trained LoRA exhibit distinct characteristics, which collectively define the overall characteristic of the trained LoRA (refer to Observation 2 in § 3.1), MOLE involves modulating the weights of different trained LoRAs within each layer, which we refer to as "hierarchical weight contro". As shown in Figure 2 (c), MOLE views each layer of trained LoRAs as a individual expert and incorporates a gating function within each layer to learn the optimal composition weights based on a specified domain objective. This dynamically enhances desirable characteristics while mitigating less favorable ones, ultimately achieving a more effective composition of LoRAs and prevents the loss of desirable LoRA characteristics that may occur in linear arithmetic composition.

Additionally, unlike reference tuning-based composition (Gu et al., 2023), our MOLE maintains flexibility in composing multiple trained LoRAs with reduced computational costs. As the workflow of MOLE shown in Figure 1, during training, MOLE learns the gating function for multiple trained LoRAs and keep all other parameters frozen, resulting in minimal computational costs. During inference, MOLE has two inference modes: In the first mode, MOLE utilizes all trained LoRAs with the learned gating function, preserving their individual characteristics with allocated weights. During the second mode, MOLE allows manual masking of unwanted LoRAs and recalculates and distributes weights proportionally without the need for retraining. These two modes enable MOLE to adapt to different scenarios, providing a versatile and flexible approach for effective LoRA composition.

We validate the effects of MOLE in both NLP and V&L domains. Our findings, encompassing both qualitative and quantitative results, demonstrate that MOLE outperforms existing LoRA composition approaches. The contributions of our paper are the following:

- We introduce a significant and intricate problem: how to dynamically and efficiently compose multiple trained LoRAs while preserving all their individual characteristics, to further investigate the applicability of LoRA in real-world scenarios.

- We introduce Mixture of LoRA Experts (MoLE), a method that achieves a more efficient and flexible composition of multiple trained LoRAs by employing hierarchical weight control through learnable gating functions within each layer of trained LoRAs.

- Extensive experiments on both V&L and NLP domain demonstrate that MoLE can enhance LoRA composition performance and mitigates issues associated with existing composition methods.

## 2 BACKGROUND

### 2.1 LoRAs COMPOSITION

LoRA (Hu et al., 2021) is a parameter-efficient fine-tuning method to adapt large models to novel tasks and shows superior performance (Hu et al., 2021; Huang et al., 2023; Zhang et al., 2023; Sung et al., 2022). In practical applications, a individual LoRA often fall short of meeting user expectations. A common solution is to compose multiple trained LoRAs, each specialized in specific aspects (e.g., clothing or facial features), with the aim of creating a comprehensive character representation. Research on LoRA composition is limited and primarily concentrates on two distinct methodologies as follows:

**Linear arithmetic composition**. As shown in Figure 2 (a), the most commonly employed composition method is directly composing multiple LoRAs, i.e.,

$$\hat{\boldsymbol{W}} = \boldsymbol{W} + \sum_{i=1}^{N} \Delta \boldsymbol{W}_i, \tag{1}$$

where $\boldsymbol{W}$ indicates the original parameter of pre-trained model and $\Delta \boldsymbol{W}_i$ denotes the $i^{th}$ trained LoRA. However, this manner may affect the original weight $\boldsymbol{W}$ when $N$ increasing, thereby diminishing the model's generative capabilities. So, it is common practice to normalize the composition weights, termed as normalized linear arithmetic composition, i.e.,

$$\hat{\boldsymbol{W}} = \boldsymbol{W} + \sum_{i=1}^{N} w_i \cdot \Delta \boldsymbol{W}_i, \tag{2}$$

where $\sum_{i=1}^{N} w_i = 1$. This manner prevents any adverse impact on the embedding of the original model, but leading to the loss of individual LoRA characteristics, as the composing weight $w_i$ for each trained LoRA is reduced (Gu et al., 2023).

In NLP domain, PEMs (Zhang et al., 2023) first define arithmetic operators for LoRA, and explore the effectiveness of composing multiple LoRAs in several scenarios. LoRAhub (Huang et al., 2023) utilizes a gradient-free manner to estimate the composition weights of trained LoRAs and achieves adaptable performance on unseen tasks. In V&L domain, SVDiff (Han et al., 2023) introduces a arithmetic-based manner to compose multiple visual concepts into a single image.

**Reference tuning-based composition**. As shown in Figure 2 (b), reference tuning-based composition (Gu et al., 2023) tackles the limitations of linear arithmetic composition by introducing gradient fusion and controllable sampling. However, it suffers from compositional inflexibility due to manually designed masks, which necessitates retraining when incorporating different LoRAs or creating new masks. Moreover, this approach entails retraining large models, resulting in substantial computational costs.

It is important to note that reference tuning-based composition relies on position masks, which distinguishes it from our model. Consequently, direct comparisons may not be appropriate due to the fundamentally different underlying principles. Therefore, our primary focus in this paper is to compare MoLE with linear arithmetic composition.

### 2.2 MIXTURE-OF-EXPERTS

Mixture-of-Experts (MoE) (Xie et al., 2023) is a promising approach to scale up the number of parameters within the same computational bounds. Different from standard transformer models, each MoE layer consists of $N$ independent feed-forward networks $\{\boldsymbol{E}_i\}_{i=0}^{N}$ as the experts, along with a

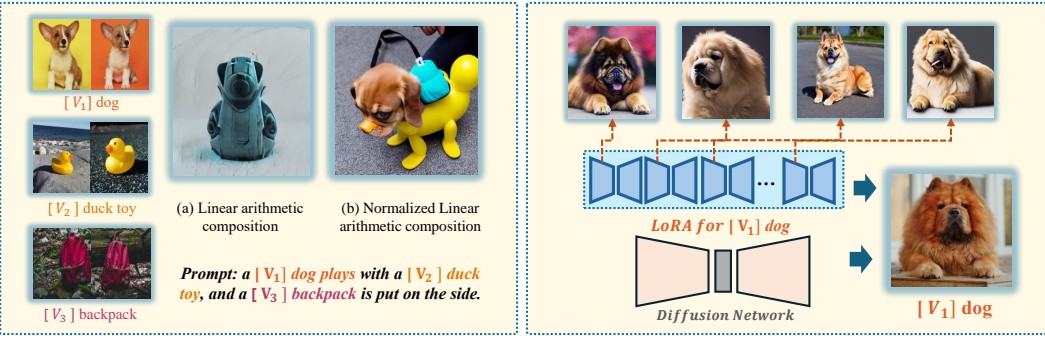

Figure 3: I. Results of (a) linear arithmetic composition (Eq. 1) and (b) normalized linear arithmetic composition (Eq. 2) based on Dreambooth (Ruiz et al., 2023). II. Visualization of the effects for different layers in LoRA by selectively activating specific parameters from the network, moving from the beginning to the end.

gating function $\alpha\left(\cdot\right)$ to model a probability distribution indicating the weights over these experts' outputs. For the hidden representation $\boldsymbol{h} \in \mathbb{R}^d$ of input token, the gate value of routing $\boldsymbol{h}$ to expert $\boldsymbol{E}_i$ is denoted as:

$$\alpha\left(\boldsymbol{E}_i\right) = \exp\left(\boldsymbol{h} \cdot \boldsymbol{e}_i\right) / \sum_{j=0}^{N} \exp\left(\boldsymbol{h} \cdot \boldsymbol{e}_j\right), \tag{3}$$

where $\boldsymbol{e}_i$ denotes the trainable embedding of $\boldsymbol{E}_i$. Then, the corresponding $k$ experts, according to the top-$k$ gated values, are activated and the output $\boldsymbol{O}$ of the MoE layer is

$$\boldsymbol{O} = \boldsymbol{h} + \sum_{i=0}^{N} \alpha\left(\boldsymbol{E}_i\right) \cdot \boldsymbol{E}_i\left(\boldsymbol{h}\right). \tag{4}$$

## 3 METHOD

In this section, we first introduce some motivating observations in § 3.1. Then, we introduce the structure details and training objectives of MoLE in § 3.2 and § 3.3, respectively.

### 3.1 MOTIVATING OBSERVATION

> **Observation 1**: *Directly composing multiple trained LoRAs (Eq. 1) impacts the model's generative ability, whereas applying weight normalization (Eq. 2) preserves this capacity but may sacrifice LoRA characteristics.*

Specifically, in V&L domain, as depicted in Figure 3 I, we observe that directly composing multiple trained LoRAs into the original embedding led to significant parameter variations, resulting in meaningless output. Furthermore, when normalization was applied, some of the original characteristics of these trained LoRAs are indeed compromised. These observations align with those elaborated upon in (Gu et al., 2023).

In NLP domain, when composing four or more LoRAs within the FLAN-T5 (Chung et al., 2022) model, we observed that the model's output became disordered. Furthermore, implementing weight normalization for LoRAs trained across five datasets, as presented in Table 4, led to a decreased performance of the composition model. This suggests that while weight normalization preserves generative capacity, it adversely affects the intrinsic qualities of these trained LoRAs.

> **Observation 2**: *Individual layers of a trained LoRA exhibit unique traits, which cumulatively define the LoRA's overall attributes.*

Inspired by the findings of (Voynov et al., 2023), which revealed that different layers in text-to-image models govern various attributes, such as style and color, we investigate the features learned

by different layers within LoRA. In V&L domain, as illustrated in Figure 3 II, we observed that different layers of LoRA encode distinct features, such as dog coat color and facial features. In NLP domain, we trained a single LoRA on a combined dataset comprising ANLI-R1 (Nie et al., 2019), ANLI-R2 (Nie et al., 2019), and QNLI (Rajpurkar et al., 2018) datasets, as depicted in Table 5. Notably, when evaluated on these sub-datasets, we observed significant variations in performance across different layers of this LoRA. Specifically, the layers ranging from 0% to 20% performed best on QNLI, the layers spanning from 40% to 60% excelled on ANLI-R2, and the layers covering 80% to 100% outperformed others on ANLI-R1. This observation inspires that we can dynamically optimizes the layer-specific weights according to a defined domain objective, enhancing desirable characteristics while suppressing less favorable ones, thereby achieving a more effective composition of trained LoRAs.

## 3.2 MIXTURE OF LORA EXPERTS

Drawing inspiration from above observations, we introduce the Mixture of LoRA Experts.

Referring to Figure 4, consider a transformer block within the pre-trained model, parameterized by $\theta$ (encompassing both the multi-head attention layer and the feed-forward neural network), and a set of corresponding trained LoRAs $\Omega = \{\Delta\theta_i\}_{i=0}^{N}$ where $N$ represents the number of trained LoRA candidates, when given a input $\boldsymbol{x} \in \mathbb{R}^{L \times d}$, the output of the pre-trained model block $\theta$ is presented as $\boldsymbol{F}_\theta \in \mathbb{R}^{L \times d}$:

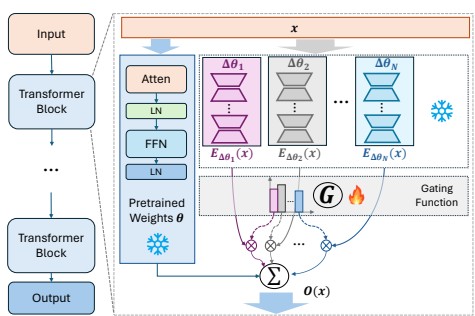

Figure 4: **Illustration of proposed MoLE.** MoLE employs a learnable gating function that utilizes the outputs of multiple LoRAs at each layer to determine composition weights.

$$\boldsymbol{x}'_\theta = \boldsymbol{x} + f_{\text{Attn}}\Big(\text{LN}\big(\boldsymbol{x}\big)\big|\theta\Big), \tag{5}$$

$$\boldsymbol{F}_\theta\big(\boldsymbol{x}\big) = \boldsymbol{x}'_\theta + f_{\text{FFN}}\Big(\text{LN}\big(\boldsymbol{x}'_\theta\big)\big|\theta\Big), \tag{6}$$

where $L$ and $d$ indicate the sequence length and the dimension of $\boldsymbol{x}$, respectively. $f_{\text{Attn}}\left(\cdot\right)$ and $f_{\text{FFN}}\left(\cdot\right)$ denotes the multi-head attention layer and feed-forward neural network, respectively. LN refers to layer normalization. The output of each LoRA is presented as $\boldsymbol{E}_{\Delta\theta_i}\left(\boldsymbol{x}\right) \in \mathbb{R}^{L \times d}$,

$$\boldsymbol{x}'_{\Delta\theta_i} = \boldsymbol{x} + f_{\text{Attn}}\Big(\text{LN}\big(\boldsymbol{x}\big)\big|\Delta\theta_i\Big), \tag{7}$$

$$\boldsymbol{E}_{\Delta\theta_i}\big(\boldsymbol{x}\big) = \boldsymbol{x}'_{\Delta\theta_i} + f_{\text{FFN}}\Big(\text{LN}\big(\boldsymbol{x}'_{\Delta\theta_i}\big)\big|\Delta\theta_i\Big). \tag{8}$$

After that, MoLE applies a learnable gating function $\mathcal{G}\left(\cdot\right)$ to model the optimal distribution of composition weights for outputs of these trained LoRAs. Specifically, by taking $\{\boldsymbol{E}_{\Delta\theta_i}\left(\boldsymbol{x}\right)\}_{i=0}^{N}$ as input, $\mathcal{G}\left(\cdot\right)$ first apply concatenation (denoted as $\oplus$) and normalization (for training stability), i.e.

$$\boldsymbol{E}_\Omega\left(\boldsymbol{x}\right) = \text{Normalization}\Big(\boldsymbol{E}_{\Delta\theta_0}\left(\boldsymbol{x}\right) \oplus \ldots \oplus \boldsymbol{E}_{\Delta\theta_{N-1}}\left(\boldsymbol{x}\right)\Big), \tag{9}$$

where $\boldsymbol{E}_\Omega\left(\boldsymbol{x}\right) \in \mathbb{R}^\xi$ and $\xi = N \times L \times d$. $\oplus$ indicates the concatenation operation. Then we flatten and reduce the $\boldsymbol{E}_\Omega\left(\boldsymbol{x}\right)$ to $N$-dimensions by a dot-product operation with the learnable parameter $\boldsymbol{e} \in \mathbb{R}^{\xi \times N}$ in the gating function $\mathcal{G}\left(\cdot\right)$,

$$\varepsilon = \text{Flatten}\Big(\boldsymbol{E}_\Omega\left(\boldsymbol{x}\right)\Big)^\top \cdot \boldsymbol{e}, \quad \varepsilon \in \mathbb{R}^N, \tag{10}$$

The gate value for each LoRA is computed as

$$\mathcal{G}\big(\varepsilon_i\big) = \frac{\exp\big(\varepsilon_i/\tau\big)}{\sum_{j=1}^{N} \exp\big(\varepsilon_j/\tau\big)}, \tag{11}$$

the temperature scalar $\tau$ is learnable. The final output $\tilde{\boldsymbol{E}}_\Omega(\boldsymbol{x})$ of the gating function $\mathcal{G}\left(\cdot\right)$ is obtained by multiplying the output of each LoRA expert with the corresponding gating values, presented as

$$\tilde{\boldsymbol{E}}_\Omega(\boldsymbol{x}) = \sum_{i=0}^{N} \mathcal{G}_i\left(\varepsilon_i\right) \cdot \boldsymbol{E}_{\Delta\theta_i}\left(\boldsymbol{x}\right), \tag{12}$$

Table 1: Text-alignment and image-alignment results for multiple LoRAs composition in CLIP feature space. NLA denotes normalized linear arithmetic composition (Eq. 2). The best performance is in bold.

| # Visual Concepts | Text-alignment | | | Image-alignment, (Concept 1) | | | Image-alignment, (Concept 2) | | | Image-alignment, (Concept 3) | | |
|---|---|---|---|---|---|---|---|---|---|---|---|---|
| | NLA | SVDiff | MoLE | NLA | SVDiff | MoLE | NLA | SVDiff | MoLE | NLA | SVDiff | MoLE |
| Fancy boot + Monster + Clock | 0.754 | 0.742 | 0.832 | 0.781 | 0.758 | 0.784 | 0.791 | 0.749 | 0.801 | 0.763 | 0.812 | 0.809 |
| Emoji + Car + Cartoon | 0.610 | 0.607 | 0.696 | 0.619 | 0.734 | 0.839 | 0.711 | 0.702 | 0.709 | 0.652 | 0.686 | 0.679 |
| Vase + Wolf plushie + Teapot | 0.752 | 0.812 | 0.863 | 0.687 | 0.807 | 0.835 | 0.705 | 0.782 | 0.746 | 0.653 | 0.694 | 0.721 |
| White Cat + Wolf plushie + Can | 0.704 | 0.772 | 0.780 | 0.801 | 0.804 | 0.802 | 0.678 | 0.763 | 0.825 | 0.650 | 0.729 | 0.714 |
| Shiny sneaker + Wolf plushie + Teapot | 0.778 | 0.789 | 0.791 | 0.812 | 0.783 | 0.690 | 0.723 | 0.751 | 0.790 | 0.688 | 0.676 | 0.721 |
| Car + Wolf plushie + Teapot | 0.635 | 0.681 | 0.684 | 0.652 | 0.763 | 0.713 | 0.601 | 0.664 | 0.745 | 0.685 | 0.612 | 0.707 |
| Can + Wolf plushie + backpack | 0.601 | 0.782 | 0.754 | 0.653 | 0.705 | 0.767 | 0.602 | 0.755 | 0.782 | 0.681 | 0.738 | 0.723 |
| Golden Retriever + Wolf plushie + Teapot | 0.670 | 0.716 | 0.784 | 0.713 | 0.784 | 0.790 | 0.601 | 0.802 | 0.809 | 0.678 | 0.761 | 0.748 |
| Golden Retriever + Boot + Monster | 0.614 | 0.762 | 0.755 | 0.665 | 0.662 | 0.620 | 0.748 | 0.832 | 0.862 | 0.723 | 0.719 | 0.735 |
| Backpack dog + Bowl + Teapot | 0.607 | 0.712 | 0.703 | 0.653 | 0.672 | 0.756 | 0.734 | 0.720 | 0.755 | 0.692 | 0.688 | 0.701 |
| Backpack dog + White Cat + Emoji | 0.648 | 0.703 | 0.717 | 0.674 | 0.692 | 0.812 | 0.719 | 0.741 | 0.701 | 0.742 | 0.720 | 0.796 |
| Dog + Wolf + Backpack | 0.717 | 0.738 | 0.722 | 0.547 | 0.565 | 0.552 | 0.679 | 0.681 | 0.707 | 0.766 | 0.795 | 0.831 |
| Cat + Sunglasses + Boot | 0.770 | 0.791 | 0.837 | 0.845 | 0.793 | 0.815 | 0.845 | 0.793 | 0.815 | 0.845 | 0.793 | 0.815 |
| Table + Can + Teapot | 0.836 | 0.827 | 0.810 | 0.753 | 0.770 | 0.741 | 0.751 | 0.799 | 0.806 | 0.818 | 0.771 | 0.829 |
| Robot + Dog + Clock | 0.663 | 0.638 | 0.693 | 0.689 | 0.764 | 0.797 | 0.645 | 0.674 | 0.710 | 0.661 | 0.715 | 0.717 |
| Average | 0.678 | 0.728 | **0.759** | 0.715 | 0.746 | **0.783** | 0.682 | 0.731 | **0.756** | 0.686 | 0.708 | **0.732** |

in which $\tilde{\boldsymbol{E}}_\Omega(\boldsymbol{x}) \in \mathbb{R}^{L \times d}$ and $\mathcal{G}_i(\cdot)$ represents the weight of the $i^{th}$ trained LoRA. So, the final output of this block is computed by adding the output of the gating function to the output of the pre-trained network:

$$\boldsymbol{O}(\boldsymbol{x}) = \boldsymbol{F}_\theta(\boldsymbol{x}) + \tilde{\boldsymbol{E}}_\Omega(\boldsymbol{x}). \tag{13}$$

Besides, we conducted an exploration of MoLE's performance when employing gating functions at different hierarchical levels (layer-wise and matrix-wise, etc). Please refer to Section 5.

### 3.3 TRAINING OBJECTIVE

**Gating Balancing Loss**. As shown in Figure 5 (a), we observed that the average entropy of the distribution probabilities from the gating functions gradually decreases as the number of training steps increases, i.e., the gating function tends to converge to a state where it always produces large weights for a early-stage well-performing LoRA (e.g., shown in Figure. 5 (b), 68% gating probability for LoRA $\beta$ among three LoRAs), leading to only a handful of LoRAs having a significant impact in the end and a loss of the characteristics of other LoRAs. To alleviate this, we propose a gating balancing loss $\mathcal{L}_{\text{balance}}$ as

$$\mathcal{L}_{\text{balance}} = -\log\left(\prod_{i=0}^{N} \mathbf{q}^{(i)}\right), \tag{14}$$

where

$$\mathbf{q}^{(i)} = \frac{1}{M} \sum_{k=1}^{M} \frac{\exp\left(\varepsilon_i^k / \tau\right)}{\sum_{j=1}^{N} \exp\left(\varepsilon_j^k / \tau\right)}, \tag{15}$$

and $M$ represents the number of blocks where gating functions are placed and $N$ denotes the number of LoRAs. This balanced loss encourages balanced gating because it is minimized when the dispatching is ideally balanced.

**Domain-specific Loss**. Additionally, for adaptation to different domains, we employ distinct domain-specific training objectives denoted as $\mathcal{L}_{\text{D}}$. In V&L domain. we employ unsupervised training with both local and global guidance from CLIP (Radford et al., 2021b) to optimize MoLE. In NLP domain, we follow the loss function in FLAN-T5 (Chung et al., 2022). The overall training objective $\mathcal{L}$ is the weighted sum of the above-mentioned two losses, represented as:

$$\mathcal{L} = \mathcal{L}_{\text{D}} + \alpha\mathcal{L}_{\text{balance}}, \tag{16}$$

where $\alpha$ is a coefficient for weight balancing.

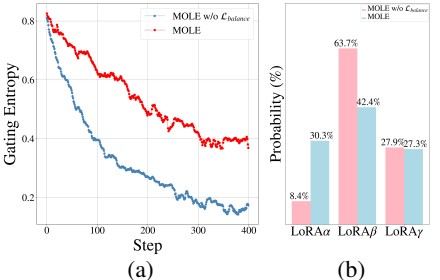

Figure 5: (a) The average gating entropy of all gating functions varies with the training steps. (b) The average weight distribution (%) of three LoRAs w and w/o $\mathcal{L}_{\text{balance}}$.

Table 2: Text-alignment and image-alignment results for multiple LoRA experts composition in CLIP feature space. The best performance is in bold and the second-best value is indicated with an underline. NLA denotes normalized linear arithmetic composition (Eq. 2). *SOTA full-parameter training methods are highlighted by*     .

| # Number of Concepts | Text-alignment | | | | | Average Image-alignment | | | | |
|---|---|---|---|---|---|---|---|---|---|---|
| | NLA | Custom | Textual Inversion | SVDiff | MoLE | NLA | Custom | Textual Inversion | SVDiff | MoLE |
| 3 | 0.678 | 0.751 | 0.709 | 0.728 | **0.759** | 0.694 | **0.761** | 0.720 | 0.719 | 0.757 |
| 4 | 0.681 | **0.735** | 0.721 | 0.717 | 0.725 | 0.712 | **0.760** | 0.736 | 0.721 | 0.742 |
| 5 | 0.652 | 0.731 | 0.704 | 0.723 | **0.762** | 0.682 | **0.798** | 0.710 | 0.708 | 0.737 |
| 6 | 0.678 | 0.722 | **0.735** | 0.709 | 0.727 | 0.698 | 0.721 | **0.747** | 0.712 | 0.736 |
| Average | 0.672 | 0.734 | 0.717 | 0.719 | **0.752** | 0.692 | **0.760** | 0.728 | 0.715 | 0.743 |

**Optimization Gating Function Only.** We freeze all trained LoRAs and pre-trained model parameters, optimizing only the gating function's parameters. This helps preserve characteristics of trained LoRAs, particularly when training data is limited.

# 4 EXPERIMENTS

## 4.1 MoLE ON V&L DOMAIN

**Experimental Setup.** For V&L domain, we apply MoLE to multi-subjects text-to-image generation task and choose DreamBooth (Ruiz et al., 2023) (built on Stable Diffusion V2.1) as the base generator. Following the common setting (Han et al., 2023; Gal et al., 2022a), where 2 to 3 concepts are typically composed into a new multi-concept image, we conduct experiments by composing three separate trained LoRAs. During training MoLE, we process the image resolution to $512 \times 512$ and set learning rate as 1e-5. We use DDPM sampler (Ho et al., 2020) with 50 steps in each case and train 400 iterations for each required composition with batch size 2 and $\alpha$ as 0.5.

**Metrics and Compared Baselines.** Following (Ruiz et al., 2023; Han et al., 2023), we evaluate our method on (1) Image alignment. The visual similarity of generated images with the individual composed concepts, using similarity in CLIP (Radford et al., 2021a) image feature space, (2) Text-alignment of the generated images with given text prompts, using text-image similarity in CLIP feature space (Radford et al., 2021a). For each composition, we calculated the average scores among 200 generated images per prompt using 5 text prompts. We compared our MoLE with normalized linear arithmetic composition (Eq. 2) and SVDiff (Han et al., 2023). Additionally, to further validate the effectiveness of MoLE, we also compare MoLE with state-of-the-art multi-subjects generation methods (full-parameters training based), which can be found in Section 5.

**Main Results.** As shown in Table 1, this study involves 15 different compositions of three visual subjects. The overall results show that our method significantly outperforms other comparative methods in terms of Text-alignment score, with a 0.031 average improvement compared to SVDiff, as well as the Image-alignment score associated with three visual concepts (e.g., 0.037 average improvement compared to SVDiff in concept 1), providing evidence of of our MoLE's superior capability in accurately capturing and depicting the subject information of user-provided images, as well as displaying multiple entities concurrently within a single image. Significantly, prior research (Kumari et al., 2023; Gal et al., 2022b) indicates a trade-off between Text-alignment and Image-alignment scores in multi-subjects generation. Excelling in both scores is challenging, highlighting the strength of our MoLE. Additionally, as shown in Figure 9, 10 and 11, our approach outperforms two other methods in preserving subject fidelity in generated images. The comparative methods often omit a subject, as seen in the NLA composition's failure to include elements like "cat" in Figure 9 (line 2) and "barn" in Figure 10, and SVDiff's inability to precisely represent "dog" and "cat" in Figure 10. Furthermore, while these methods can generate images with three subjects, there's a noticeable leakage and mixing of appearance features, resulting in lower subject fidelity compared to user-provided images. In contrast, our method effectively retains the subjects specified by the user, with each accurately depicted.

## 4.2 MoLE ON NLP DOMAIN

**Experimental Setup.** For NLP domain, following (Huang et al., 2023), we employ Flan-T5 (Chung et al., 2022) as our chosen LLM and created several LoRAs based on FLAN datasets. We conducted

extensive experiments across various tasks, including Translation, Natural Language Inference (NLI), Struct to Text, Closed-Book QA, and multiple subtasks within the Big-Bench Hard (BBH) (Ghazal et al., 2013) dataset. We train 800 iterations for each required composition of LoRAs with an initial learning rate of 1e-5, batch size 12 and $\alpha$ as 0.5.

**Compared Baselines.** We compared our MOLE with recently released state-of-the-art LoRA composition methods: LoRAhub (Han et al., 2023) and PEMs (Zhang et al., 2023).

**Main Results.** The corresponding experimental results are encapsulated in the Table 3. In summary, our MOLE surpasses state-of-the-art LoRA composition methods on five distinct datasets. Notably, on the BBH dataset, our MOLE achieves an average performance improvement of 3.8 over LoRAHub and outperforms PEMs by a notable margin of 9.0. Furthermore, in the realm of generation tasks, specifically in Translation and Struct to Text categories, MOLE consistently outshines its counterparts. In the Translation task set, it surpasses LoRAHub by an average margin of 1.5 and PEMs by 2.7. Correspondingly, within the Struct to Text task set, our model boasts an average performance superiority of 2.1 over LoRAHub and 2.6 over PEMs. These findings underscore the efficacy and versatility of our MOLE in handling language generation tasks.

| # Task | Metric | LoRAHub | PEMs | MoLE |
|---|---|---|---|---|
| **Translation** | | | | |
| WMT '14 En→Fr | BLEU | 27.4 | 25.6 | **29.1** |
| WMT '14 Fr→En | BLEU | 29.4 | 27.1 | **31.3** |
| WMT '16 En→De | BLEU | 24.6 | 24.9 | **27.7** |
| WMT '16 De→En | BLEU | **29.9** | 28.0 | 29.1 |
| WMT '16 En→Ro | BLEU | 17.7 | 15.2 | **18.9** |
| WMT '16 Ro→En | BLEU | 23.5 | 21.7 | **25.1** |
| Average | | 25.4 | 24.2 | **26.9** |
| **Struct to Text** | | | | |
| CommonGen | Rouge-1 | 53.7 | 48.8 | **55.1** |
| | Rouge-2 | **23.1** | 22.4 | 23.1 |
| | Rouge-L | 49.7 | 47.2 | **53.9** |
| DART | Rouge-1 | 45.3 | 46.2 | **48.8** |
| | Rouge-2 | 22.6 | 18.9 | **23.5** |
| | Rouge-L | 35.1 | **37.6** | 36.0 |
| E2ENLG | Rouge-1 | 41.1 | 40.7 | **42.0** |
| | Rouge-2 | 26.3 | 24.2 | **29.0** |
| | Rouge-L | 38.8 | **42.1** | 41.8 |
| WebNLG | Rouge-1 | 52.1 | 52.0 | **54.5** |
| | Rouge-2 | 23.9 | 24.6 | **26.8** |
| | Rouge-L | 45.2 | 47.8 | **49.3** |
| Average | | 38.1 | 37.7 | **40.3** |
| **Closed-Book QA** | | | | |
| ARC-c | EM | 51.7 | 50.4 | **52.9** |
| ARC-e | EM | 69.7 | 65.7 | **70.3** |
| NQ | EM | 17.3 | 16.1 | **23.5** |
| TQA | EM | **54.5** | 53.9 | 54.0 |
| Average | | 48.3 | 46.5 | **50.2** |
| **Big-Bench Hard (BBH)** | | | | |
| Boolean Expressions | EM | 55.1 | 53.0 | **57.3** |
| Causal Judgement | EM | 57.6 | 51.1 | **57.9** |
| Date Understanding | EM | **31.0** | 29.3 | 30.7 |
| Disambiguation | EM | 46.6 | 47.2 | **49.3** |
| Penguins in a Table | EM | 41.4 | 39.8 | **45.0** |
| Reasoning Objects | EM | 35.2 | **37.5** | 33.7 |
| Ruin Names | EM | 19.9 | 19.3 | **21.2** |
| Average | | 38.4 | 33.2 | **42.2** |
| **Natural Language Inference (NLI)** | | | | |
| ANLI-R1 | EM | 81.0 | 80.3 | **82.7** |
| ANLI-R2 | EM | 80.9 | 80.2 | **82.4** |
| ANLI-R3 | EM | 77.4 | 76.6 | **78.9** |
| QNLI | EM | 77.6 | 78.0 | **78.1** |
| Average | | 79.2 | 78.8 | **80.5** |

Table 3: Evaluation results on Translation, Struct to Text, Closed-Book QA, NLI and BBH. The **best value** is in bold and the second-best value is underlined.

## 5 ANALYSIS

**The effectiveness of gating balancing loss**. Figure 5 (a) and (b) illustrate how our $\mathcal{L}_{balance}$ function mitigates the reduction in entropy rates within gating functions, leading to a more uniform composition weight distribution. The performance comparison between MOLE and MOLE $_{w/o\ \mathcal{L}_{balance}}$ in Table 7 underscores the performance enhancement achieved with the inclusion of $\mathcal{L}_{balance}$. Additionally, we conducted an experiment wherein we solely increased the temperature $\tau$ in Eq. 11, as an alternative to adding $\mathcal{L}_{balance}$. Results in Table 7 shows declining performance in MOLE variants MOLE$^{\tau_1}$, MOLE$^{\tau_2}$, MOLE$^{\tau_3}$ ($\tau_1 \prec \tau_2 \prec \tau_3$) with increasing temperature. While temperature rise addresses gating imbalance, it restricts dynamic LoRA exploration in MOLE, leading to inferior outcomes.

**Further comparison with SOTA multi-concept generation methods**. In the absence of comparable LoRA composition methods in the V&L domain, we incorporated two leading multi-concept generation algorithms that do not utilize LoRA: Custom (Kumari et al., 2023) and Textual Inversion (Gal et al., 2022a), both of which emphasize full-parameter training for enhanced results. As presented in Table 2, MOLE outperforms Textual Inversion in both image and text alignment and excels over Custom in text alignment. Furthermore, it's worth noting that our MoLE is more lightweight compared to these full-parameter training methods. These comparisons underscore the superior effectiveness of our MoLE relative to methods that involve extensive parameter tuning.

**Scale to a larger number of LoRAs**. We explore the performance as the number of LoRAs increases. In the NLP domain, experiments were conducted with varying numbers of LoRA (8, 24, 48, 128),

as detailed in Table 6. Our MOLE demonstrated optimal performance across these configurations, notably excelling with larger LoRA counts of 48 and 128, surpassing LoRAHub by **2.5** and **3.0**, respectively. Analysis revealed that LoRAHub's optimization algorithm often zeroes out many LoRA weights in larger arrays, thus underutilizing the potential of all LoRA. Conversely, MOLE effectively overcomes this limitation. However, all methods, including MOLE, showed performance declines with an extremely large number of LoRA (128), highlighting a need for further research in this area. In the V&L domain, Table 10 shows experiments with increased composed LoRAs. While typical composition involve 3-4 visual concepts, our range was 3-6 to avoid ambiguity in outputs. Results indicate that MOLE consistently outperforms other LoRA composition models in text and image alignment as the number of LoRAs increases, underscoring its robustness and superior composition capabilities.

**Coarse-to-fine gating analysis**. To examine the impact of different granularity levels in gating functions, we delineated four levels in MOLE: matrix-wise (MOLE, gating at the parameter matrix level), layer-wise (MOLE), block-wise (MOLE), and network-wise (MOLE), abbreviated as m-MOLE, l-MOLE, b-MOLE, and n-MOLE respectively. Table 9 reveals that intermediate granularities, b-MOLE and l-MOLE, achieved the highest performance. In contrast, the coarsest level, n-MOLE, which involves minimal optimizable parameters (a single gating for the entire network), showed suboptimal outcomes. Additionally, the finest granularity, m-MOLE, underperformed, potentially due to its excessive control interfering with inherent relationships in LoRA parameters.

**Generalization to new datasets**. To further validate the effectiveness of our MOLE, we conducted generalization experiments. Specifically, all LoRA candidates and LoRA composition variants, including MOLE, PEMs and LoRAHub, were trained on NLI tasks (ANLI-R1, ANLI-R2, ANLI-R3, QNLI, and WNLI, among others). Subsequently, we evaluated these methods on the BBH dataset. As illustrated in Table 8, our MOLE achieves an average performance advantage of 2.4 over LoRAHub and 3.7 over PEMs, underscoring its superior generalization ability.

**Flexibility of MOLE**. As discussed in Section 2.1, a well-designed LoRA composition method should not only achieve effective LoRA composition but also retain the characteristics of individual LoRA. It should be versatile enough to function as a standalone LoRA generator, ensuring its practical applications are flexible and widespread. Figure 6 displays a comparison of the qualitative results for the retaining ability of several composition methods, we find that our MOLE can generate images that closely resemble the original features of the LoRA experts (e.g., dog ears, the color of the backpack), while other composition methods tend to produce confusion and loss of LoRA characteristics. Besides, as shown in Figure 1, we can also degrade MOLE by masking out the LoRA experts we do not wish to use, transforming it into a MOLE that merges fewer LoRAs without affecting the composition effect of the remaining LoRAs. As shown in Figure 8, our MOLE can achieve the same flexible LoRA composition as linear arithmetic composition method without altering the weights of MOLE, while reference tuning-based composition (Gu et al., 2023) can not accomplish.

**Hierarchical control analysis**. MOLE aims to achieve improved LoRA composition effects through finer-grained hierarchical control. As illustrated in the Figure 7, we visualize the weight distributions assigned by the gating functions learned by MOLE at different levels in both NLP and V&L domains. We observe that MOLE adaptively assigns weights to different LoRA experts at various layers. Consequently, finer-grained weight combination methods lead to superior results.

# 6    CONCLUSION AND LIMITATIONS

In this study, we introduce the Mixture of LoRA Experts (MOLE) as a versatile and dynamic approach for composing multiple trained LoRAs. The key innovation of MOLE lies in its learnable gating functions, which utilize the outputs of multiple LoRAs at each layer to determine composition weights. Our comprehensive evaluation in both the both NLP and V&L domains establishes that MOLE outperforms existing LoRA composition methods.

**Limitations**. As described in Section 5, when the number of LoRAs increases to a very large value (e.g., 128), despite our MOLE exhibiting superior performance, the performance of all LoRA composition methods, including our MOLE, tends to decrease. This suggests that our MOLE still faces challenges when performing large-scale LoRA composition. It also highlights the significance of researching better approaches for handling large-scale LoRA composition effectively.

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

Table 4: The first motivation experiment in the NLP domain. NLA denotes normalized linear arithmetic composition (Eq. 2). The **best value** is in bold.

| Model | ANLI-R1 | ANLI-R2 | ANLI-R3 | QNLI | WNLI | Average |
|---|---|---|---|---|---|---|
| Single LoRA | **80.32** | **79.02** | 75.92 | **78.62** | **74.32** | **77.64** |
| NLA | 79.32 | 78.88 | **76.42** | 78.06 | 69.98 | 76.53 |

Table 5: The second motivation experiment in the NLP domain. Full LoRA denotes the application of the complete set of LoRA parameters for inference, whereas x%-y% indicates the inference using LoRA parameters ranging from the top x% to the top y%. The **best value** is in bold.

| | ANLI-R1 | ANLI-R2 | QNLI |
|---|---|---|---|
| Full LoRA | 81.65 | 80.03 | 76.42 |
| 0%-20% | 78.72 | 78.35 | **78.14** |
| 20%-40% | 76.10 | 77.96 | 77.85 |
| 40%-60% | 76.95 | **81.47** | 74.57 |
| 60%-80% | 77.25 | 78.19 | 75.71 |
| 80%-100% | **82.59** | 77.91 | 75.48 |

Table 6: NLP domain experimental results on the impact of exploring expand expert numbers on model performance. The result is the average EM on the Big-Bench Hard (BBH) dataset. NLA denotes normalized linear arithmetic composition (Eq. 2). The **best value** is in bold and the second-best value is indicated with an underline.

| # Number of LoRA | NLA | LoRAHub | PEMs | MoLE |
|---|---|---|---|---|
| 8 | 32.7 | 33.9 | 33.7 | **36.6** |
| 24 | 36.8 | 37.1 | 36.9 | **38.7** |
| 48 | 34.4 | 36.9 | 34.6 | **39.4** |
| 128 | 34.1 | 35.5 | 34.9 | **38.5** |
| Average | 34.5 | 35.9 | 35.0 | **38.3** |

Table 7: Experimental results on gating balance of MoLE. NLA denotes normalized linear arithmetic composition (Eq. 2). The **best value** is in bold.

| # Model | ANLI-R1 | ANLI-R2 | ANLI-R3 | QNLI | WNLI | Average |
|---|---|---|---|---|---|---|
| NLA | 79.32 | 78.88 | 76.42 | 78.06 | 69.98 | 76.53 |
| MoLE | **81.49** | **79.38** | **77.63** | **79.52** | **72.31** | **78.07** |
| MoLE w/o $\mathcal{L}_{\text{balance}}$ | 80.81 | 79.11 | 77.42 | 79.09 | 71.44 | 77.57 |
| MoLE$^{\tau_1}$ | 80.52 | 79.27 | 77.30 | 79.11 | 71.07 | 77.45 |
| MoLE$^{\tau_2}$ | 80.01 | 79.03 | 76.33 | 77.81 | 70.37 | 76.71 |
| MoLE$^{\tau_3}$ | 78.50 | 79.20 | 76.07 | 78.02 | 70.00 | 76.35 |

Table 8: Evaluation results on generalization to new datasets. All lora candidates and LoRA merging variants are optimized on NLI tasks. The **best value** is in bold and the second-best value is indicated with an underline.

| # Task | Metric | LoRAHub | PEMs | MoLE |
|---|---|---|---|---|
| **Big-Bench Hard (BBH)** | | | | |
| Boolean Expressions | EM | 45.3 | 45.5 | **48.7** |
| Causal Judgement | EM | 51.3 | 46.1 | **52.4** |
| Date Understanding | EM | **27.5** | 24.6 | 26.6 |
| Disambiguation | EM | 39.7 | 42.4 | **43.8** |
| Penguins in a Table | EM | 35.3 | 33.6 | **39.0** |
| Reasoning about Colored Objects | EM | 32.2 | 31.4 | **34.7** |
| Average | | 38.5 | 37.2 | **40.9** |

Table 9: Coarse-to-fine gating comparison. The **best value** is in bold and the second-best value is indicated with an underline.

| # Method | Text-alignment | Image-alignment | | |
|---|---|---|---|---|
| | | Concept 1 | Concept 2 | Concept 3 |
| m-MoLE | 0.731 | 0.719 | 0.714 | 0.747 |
| l-MoLE | 0.760 | 0.727 | 0.731 | **0.757** |
| b-MoLE | **0.766** | 0.726 | **0.737** | 0.755 |
| n-MoLE | 0.722 | **0.739** | 0.682 | 0.730 |

Table 10: Experimental results on the impact of exploring expand expert numbers on model performance. We evaluate each composition pair on 200 images generated using 5 prompts with 50 steps of DDPM sampler and scale=7.5. NLA denotes normalized linear arithmetic composition (Eq. 2). The best performance is in bold.

| # Number of LoRA | Text-alignment | | | Average Image-alignment | | |
|---|---|---|---|---|---|---|
| | NLA | SVDiff | MoLE | NLA | SVDiff | MoLE |
| 3 | 0.678 | 0.728 | **0.759** | 0.694 | 0.719 | **0.757** |
| 4 | 0.681 | 0.717 | **0.725** | 0.712 | 0.721 | **0.742** |
| 5 | 0.652 | 0.723 | **0.762** | 0.682 | 0.708 | **0.737** |
| 6 | 0.698 | 0.709 | **0.737** | 0.703 | 0.701 | **0.709** |
| Average | 0.677 | 0.719 | **0.746** | 0.698 | 0.712 | **0.736** |

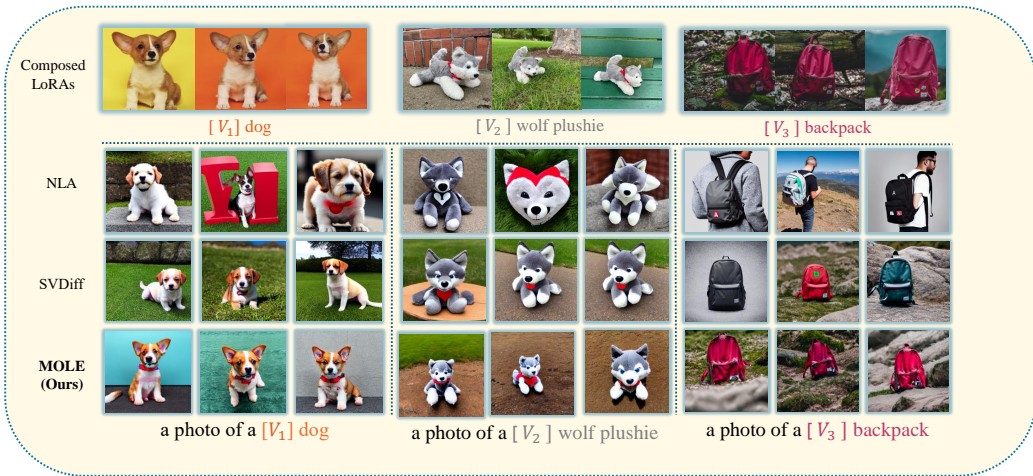

Figure 6: Qualitative result for retaining ability experiment. NLA denotes normalized linear arithmetic composition (Eq. 2). The first row displays the composed trained LoRAs. The second to the last row showcases the respective abilities of different composition methods to preserve the characteristics of each LoRA without altering the model.

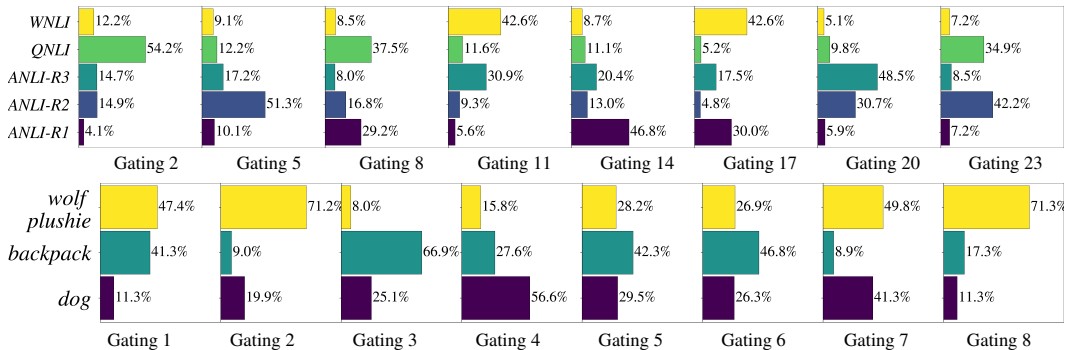

Figure 7: Visualization of the weights (%) predicted by each gating function (horizontal axis) for LoRA experts (vertical axis) during inference. The top row corresponds to experiments in the NLP domain, while the bottom row pertains to experiments in the V&L domain.

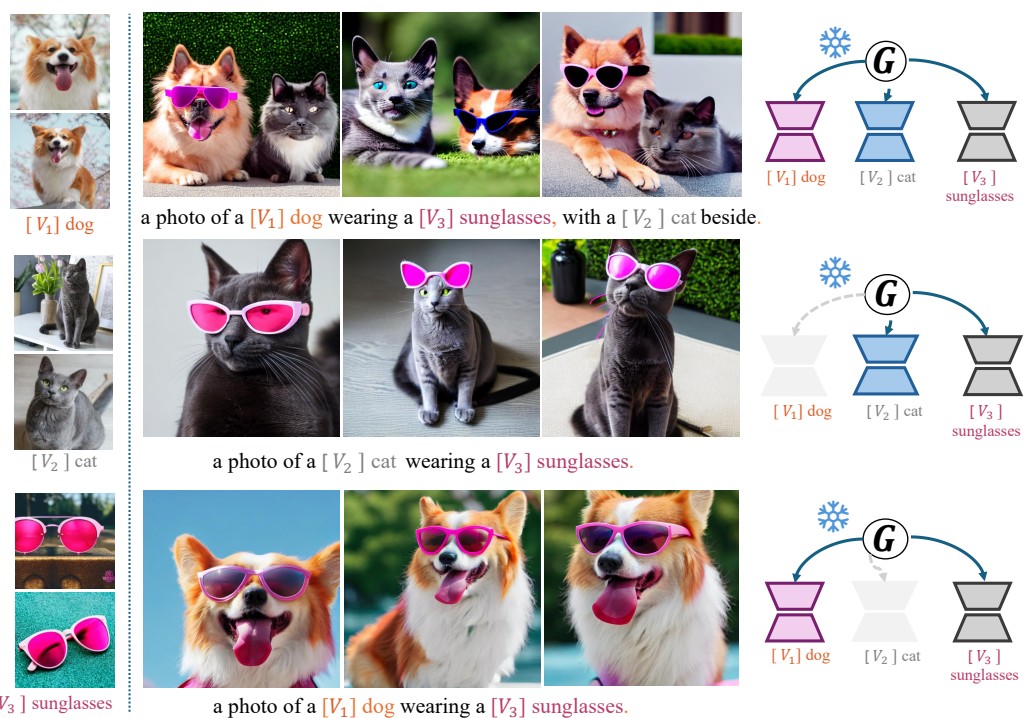

Figure 8: Visualization for different inference modes of MOLE. MOLE has two inference modes: In the first mode (the first line), MOLE can use all the LoRA experts and allocate weights for each LoRA, preserving their individual characteristics. In the second mode (the second and third lines), we can manually mask some unwanted LoRAs without changing the gating weights. It can recalculate and distribute weights proportionally. These two modes enable MOLE to adapt to different scenarios, providing a versatile and flexible approach for effective LoRA composition.

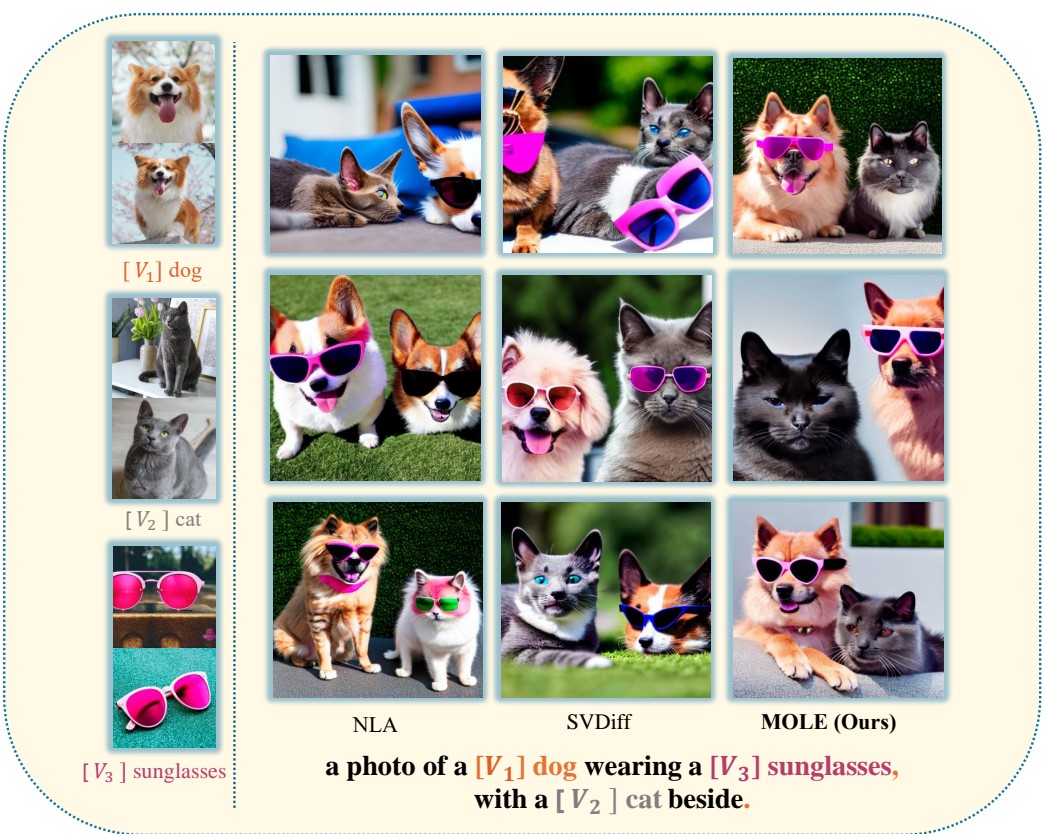

Figure 9: Visualization of multiple LoRA composition results on V&L domain. NLA denotes normalized linear arithmetic composition (Eq. 2). Our MoLE has higher visual similarity with the personal cat and dog images while following the text condition better, e.g., SVDiff is unable to fully recover all the characteristics of LoRA (in the second line, the appearance of the dog is completely altered, and in the first line, two cats are present but the dog is missing). Moreover, SVDiff and NLA struggles to generate images that match the text condition effectively (e.g., it might add sunglasses to both dogs and cats in response to conditions mentioning "dog" and "cat").

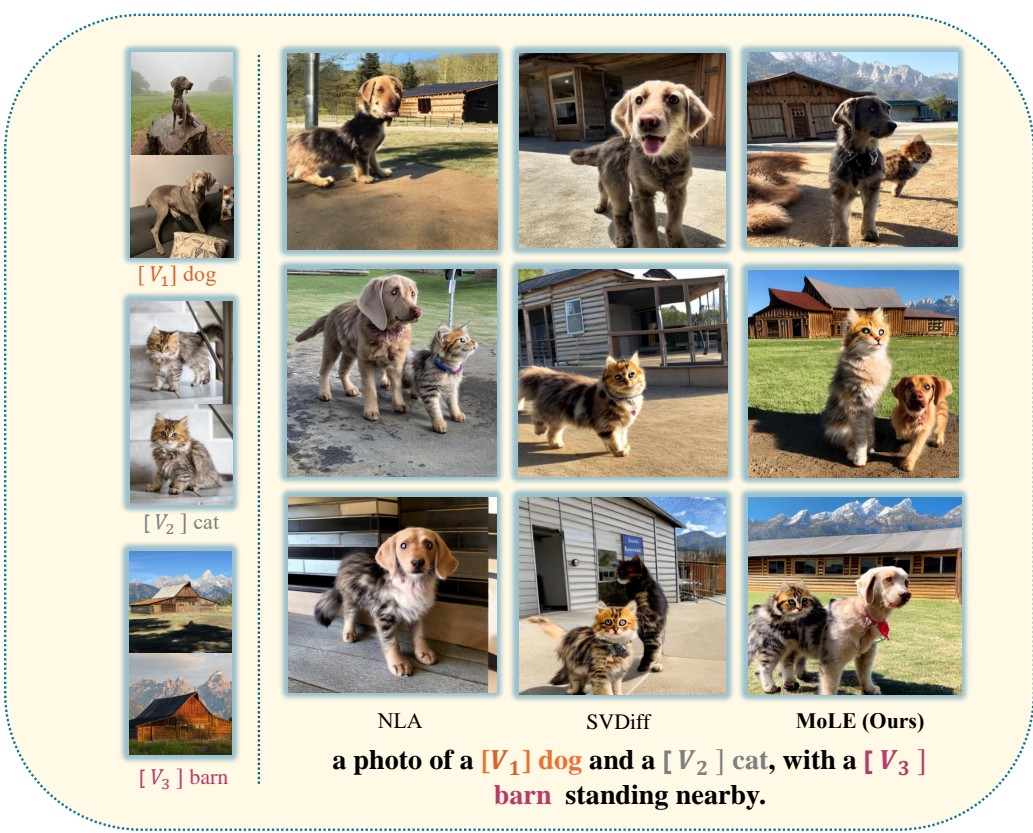

Figure 10: Visualization of multiple LoRA composition results on V&L domain. NLA denotes normalized linear arithmetic composition (Eq. 2). Our model consistently produces results that better align with the prompt descriptions. The outputs from our model consistently contain all three visual concepts that need to be combined. In contrast, SVDiff and NLA often exhibit issues such as concept confusion (e.g., in the third row of NLA, where features of both the cat and dog are confused) and concept omission (e.g., in the second row of SVDiff, where the concept of the dog is missing, and in the first row, where the concept of the cat is missing).

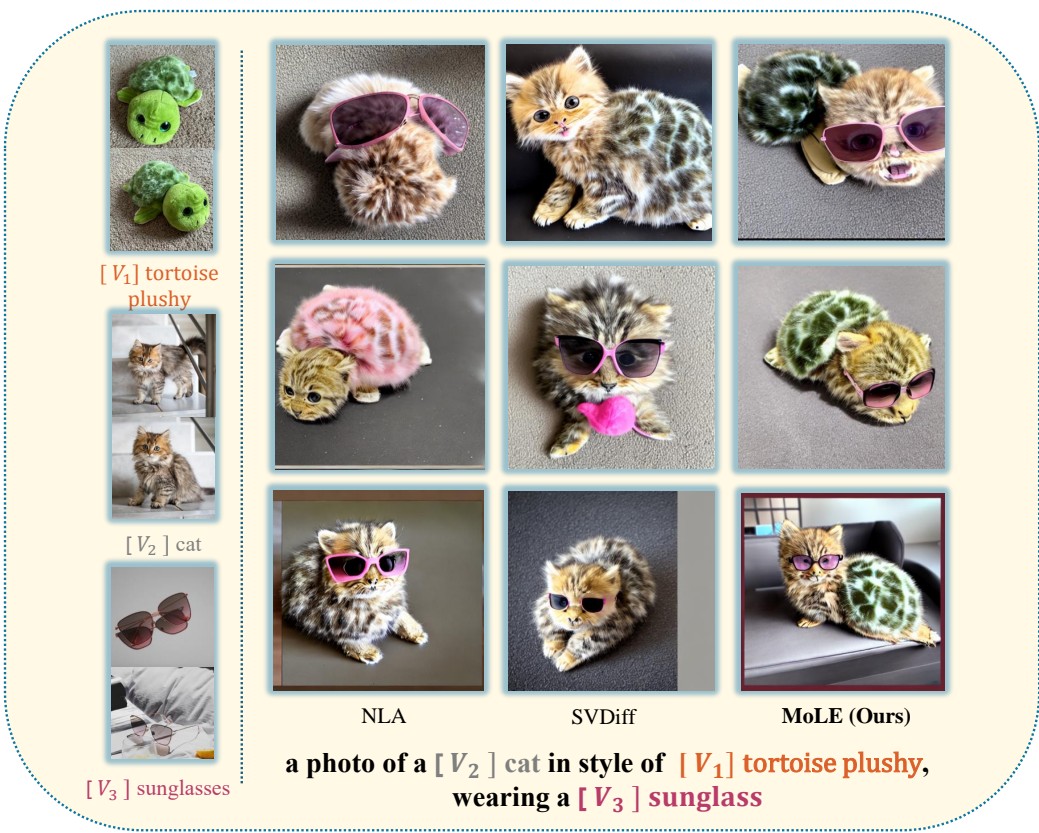

Figure 11: Visualization of multiple LoRA composition results on V&L domain. NLA denotes normalized linear arithmetic composition (Eq. 2). Our model consistently produces results that better align with the prompt descriptions. The outputs from our model consistently contain all three visual concept features that need to be combined. In contrast, SVDiff and NLA often exhibit issues such as concept omission (e.g., in the first row of NLA, where the concepts of the cat and sunglasses are missing, and in the first row of SVDiff, where the concept of sunglasses is missing). Additionally, our output results better match the original visual concept features. For example, the shell of the turtle is green, whereas SVDiff and NLA generate shells in pink and brown colors.

