# OpenReview forum: "Mixture of LoRA Experts"
_ICLR.cc/2024/Conference — ICLR 2024 poster_

### Official Review · Reviewer_73Bx · 2023-10-25

**Soundness:** 2 fair
**Presentation:** 2 fair
**Contribution:** 2 fair
**Rating:** 3
**Confidence:** 4

**Summary:**

This paper introduces the Mixture of LoRA Experts (MOLE), a learnable gating function for combining multiple LoRAs. MOLE provides fine-grained hierarchical control. Gating balancing loss is used to train the learnable gate parameter. It surpasses the performance of direct arithmetic merging and retains the necessary flexibility to combine LoRAs effectively.

**Strengths:**

* The proposed method is simple and the architecture is light-weighted.

* The authors define the problems reasonably and test the MoLE in two significant domains, NLP and V&L, showcasing its versatility.

**Weaknesses:**

* The generation of images still meets problems. For example, the generated images do not match the text condition effectively, as shown in the first row of Figure 9. Most importantly, in the second and third rows, the texture of the V1 dog generated by MoLE is not well-preserved compared to the texture generated by other baselines, especially to simple fusion.

* The samples are not enough to support the claims in VL tasks. Only one example demonstrates the ability to combine multiple experts in Figure 9, and only three examples are shown for a single-expert combination. Most importantly, only four examples are used for quantitative experiments, which is insufficient to get solid conclusions.

* The improvement is marginal for both VL and NLP tasks. From Table 1, we can find that MOLE cannot achieve the best text-alignment score for 2/4 cases and cannot achieve the best image-alignment score in the second row for 3/4 cases. For NLP tasks, the improvement is marginal in many cases of cross-domain and in-domain settings. Hence, more experiments are needed to demonstrate the effectiveness of the proposed MoLE.

**Questions:**

* Could you provide more examples for both quantitative and qualitative experiments of VL tasks?

* Could you provide more results on different NLP datasets and tasks?

---

> ### Author Response · Authors · 2023-11-21
> **Rebuttal from Authors of Paper4944 to Reviewer 73Bx (1/3)**
>
> We thank the Reviewer for the insightful suggestions. Regarding the questions raised:
>
> -------
> > **Q1**: Could you provide more results on different NLP datasets and tasks?
>
> > **Q2**: For NLP tasks, the improvement is marginal in many cases of cross-domain and in-domain settings. Hence, more experiments are needed to demonstrate the effectiveness of the proposed MoLE.
>
> **A1 & A2**. We conducted extensive experiments across various tasks, including **Translation**, **Struct to Text**, **Closed-Book QA**, and multiple subtasks within the challenging **Big-Bench Hard (BBH)** datasets. Additionally, we introduced a new LoRA merging variant called **PEMs** $^{[1]}$, recently proposed in the field of NLP.
>
> The corresponding experimental results are summarized in the following table. In summary, our MoLE surpasses state-of-the-art LoRA merging variants on four distinct datasets and tasks, showcasing robust performance. With a notable highlight on the BBH dataset, our MoLE achieves an average performance improvement of **3.8** over LoRAHub and outperforms PEMs by a substantial margin of **9.0**.
>
> In the domain of natural language generation tasks (**Translation** and **Struct to Text**), our MoLE consistently demonstrates superior average performance in the Translation task set, surpassing LoRAHub by **1.5** and PEMs by **2.7**. Similarly, within the Struct to Text task set, our model achieves an average performance advantage of **2.1** over LoRAHub and **2.6** over PEMs. These results highlight the effectiveness of our model in generation task.
>
>
>
> | Task                            | Metric  | LoRAHub  | PEMs     | Our      |
> | ------------------------------- | ------- | -------- | -------- | -------- |
> | **Translation**                 |         |          |          |          |
> | WMT '14 En->Fr                  | BLEU    | 27.4     | 25.6     | **29.1** |
> | WMT '14 Fr->En                  | BLEU    | 29.4     | 27.1     | **31.3** |
> | WMT '16 En->De                  | BLEU    | 24.6     | 24.9     | **27.7** |
> | WMT '16 De->En                  | BLEU    | **29.9** | 28       | 29.1     |
> | WMT '16 En->Ro                  | BLEU    | 17.7     | 15.2     | **18.9** |
> | WMT '16 Ro->En                  | BLEU    | 23.5     | 21.7     | **25.1** |
> | Average                         | BLEU    | 25.4     | 24.2     | **26.9** |
> | **Struct to Text**              |         |          |          |          |
> | CommonGen                       | Rouge-1 | 53.7     | 48.8     | **55.1** |
> |                                 | Rouge-2 | **23.1** | 22.4     | 23.1     |
> |                                 | Rouge-L | 49.7     | 47.2     | **53.9** |
> | DART                            | Rouge-1 | 45.3     | 46.2     | **48.8** |
> |                                 | Rouge-2 | 22.6     | 18.9     | **23.5** |
> |                                 | Rouge-L | 35.1     | **37.6** | 36.0     |
> | E2ENLG                          | Rouge-1 | 41.1     | 40.7     | **42.0** |
> |                                 | Rouge-2 | 26.3     | 24.2     | **29.0** |
> |                                 | Rouge-L | 38.8     | **42.1** | 41.8     |
> | WebNLG                          | Rouge-1 | 52.1     | 52.0     | **54.5** |
> |                                 | Rouge-2 | 23.9     | 24.6     | **26.8** |
> |                                 | Rouge-L | 45.2     | **47.8** | 49.3     |
> | Average                         |         | 38.1     | 37.7     | **40.3** |
> | **Closed-Book QA**              |         |          |          |          |
> | ARC-c                           | EM      | 51.7     | 50.4     | **52.9** |
> | ARC-e                           | EM      | 69.7     | 65.7     | **70.3** |
> | NQ                              | EM      | 17.3     | 16.1     | **23.5** |
> | TQA                             | EM      | **54.5** | 53.9     | 54.0     |
> | Average                         | EM      | 48.3     | 46.5     | **50.2** |
> | **Big-Bench Hard (BBH)**        |         |          |          |          |
> | Boolean Expressions             | EM      | 55.1     | 53.0     | **57.3** |
> | Causal Judgement                | EM      | 57.6     | 51.1     | **57.9** |
> | Date Understanding              | EM      | **31.0** | 29.3     | 30.7     |
> | Disambiguation                  | EM      | 46.6     | 47.2     | **49.3** |
> | Penguins in a Table             | EM      | 41.4     | 39.8     | **45.0** |
> | Reasoning about Colored Objects | EM      | 35.2     | **37.5** | 33.7     |
> | Ruin Names                      | EM      | 19.9     | 19.3     | **21.2** |
> | Average                         | EM      | 38.4     | 33.2     | **42.2** |
>
> **References**
>
> [1] Zhang, J., Chen, S., Liu, J., & He, J. (2023). Composing parameter-efficient modules with arithmetic operations. arXiv preprint arXiv:2306.14870.

---

> ### Author Response · Authors · 2023-11-21
> **Rebuttal from Authors of Paper4944 to Reviewer 73Bx (2/3)**
>
> ----------------------
> > **Q3**: The improvement is marginal for both VL and NLP tasks. From Table 1, we can find that MOLE cannot achieve the best text-alignment score for 2/4 cases and cannot achieve the best image-alignment score in the second row for 3/4 cases.
>
> > **Q4**: The samples are not enough to support the claims in VL tasks.
>
> **A3 &  A4**. As shown in following Table, there is usually a trade-off between text-alignment and image-alignment$^{[1][2]}$ on the multi-concepts fusion in V & L domain (The visualization of this trade-off can be seen in the Figure.12 in the supp). High image-alignment leads to a decrease in text-alignment where sample generations have less variance and are close to input target images. As a result, our MoLE may not perform better in both image and text sub alignment.
>
> | Training Steps | Text Alignment Similarity | Image Alignment Similarity |
> | -------------- | ------------------------- | -------------------------- |
> | 500            | 0.83                      | 0.68                       |
> | 1000           | 0.81                      | 0.73                       |
> | 1500           | 0.80                      | 0.75                       |
> | 2000           | 0.795                     | 0.76                       |
> | 2500           | 0.792                     | 0.77                       |
> | 3000           | 0.786                     | 0.774                      |
>
> However, as depicted in Table 1 in the main text, it is worth noting that when evaluating the overall average across all configurations, our MoLE consistently outperforms both in average text alignment and image alignment. This underscores MoLE's capability to achieve superior alignment between text and images. Furthermore, we present additional results in the V&L domain following to further validate MoLE's effectiveness. In these results, it's evident that MoLE achieves the highest average scores in both image alignment and text alignment.
>
> | Number of LoRA | Text-alignment (Simple Fusion) | Text-alignment (SVDiff) | Text-alignment (MoLE) | Image-alignment (LoRA expert 1, Simple Fusion) | Image-alignment (LoRA expert 1, SVDiff) | Image-alignment (LoRA expert 1, MoLE) | Image-alignment (LoRA expert 2, Simple Fusion) | Image-alignment (LoRA expert 2, SVDiff) | Image-alignment (LoRA expert 2, MoLE) | Image-alignment (LoRA expert 3, Simple Fusion) | Image-alignment (LoRA expert 3, SVDiff) | Image-alignment (LoRA expert 3, MoLE) |
> |-----------------|-----------------------------|-------------------------|---------------------|--------------------------------------------|------------------------------------------|-------------------------------------|--------------------------------------------|------------------------------------------|-------------------------------------|--------------------------------------------|------------------------------------------|-------------------------------------|
> | Fancy boot + Monster + Clock | 0.754 | 0.742 | 0.832 | 0.781 | 0.758 | 0.784 | 0.791 | 0.749 | 0.801 | 0.763 | 0.812 | 0.809 |
> | Emoji + Car + Cartoon | 0.610 | 0.607 | 0.696 | 0.619 | 0.734 | 0.839 | 0.711 | 0.702 | 0.709 | 0.652 | 0.686 | 0.679 |
> | Vase + Wolf plushie + Teapot | 0.752 | 0.812 | 0.863 | 0.687 | 0.807 | 0.835 | 0.705 | 0.782 | 0.746 | 0.653 | 0.694 | 0.721 |
> | White Cat + Wolf plushie + Can | 0.704 | 0.772 | 0.780 | 0.801 | 0.804 | 0.802 | 0.678 | 0.763 | 0.825 | 0.650 | 0.729 | 0.714 |
> | Shiny sneaker + Wolf plushie + Teapot | 0.778 | 0.789 | 0.791 | 0.812 | 0.783 | 0.690 | 0.723 | 0.751 | 0.790 | 0.688 | 0.676 | 0.721 |
> | Car + Wolf plushie + Teapot | 0.635 | 0.681 | 0.684 | 0.652 | 0.763 | 0.713 | 0.601 | 0.664 | 0.745 | 0.685 | 0.612 | 0.707 |
> | Can + Wolf plushie + backpack | 0.601 | 0.782 | 0.754 | 0.653 | 0.705 | 0.767 | 0.602 | 0.755 | 0.782 | 0.681 | 0.738 | 0.723 |
> | Golden Retriever + Wolf plushie + Teapot | 0.670 | 0.716 | 0.784 | 0.713 | 0.784 | 0.790 | 0.601 | 0.802 | 0.809 | 0.678 | 0.761 | 0.748 |
> | Golden Retriever + Boot + Monster | 0.614 | 0.762 | 0.755 | 0.665 | 0.662 | 0.620 | 0.748 | 0.832 | 0.862 | 0.723 | 0.719 | 0.735 |
> | Backpack dog + Bowl + Teapot | 0.607 | 0.712 | 0.703 | 0.653 | 0.672 | 0.756 | 0.734 | 0.720 | 0.755 | 0.692 | 0.688 | 0.701 |
> | Backpack dog + White Cat + Emoji | 0.648 | 0.703 | 0.717 | 0.674 | 0.692 | 0.812 | 0.719 | 0.741 | 0.701 | 0.742 | 0.720 | 0.796 |
> | Average | 0.678 | 0.728 | **0.759** | 0.715 | 0.746 | **0.783** | 0.682 | 0.731 | **0.756** | 0.686 | 0.708 | **0.732** |
>
> **References**
>
> [1] Kumari, N., Zhang, B., Zhang, R., Shechtman, E., & Zhu, J. Y. (2023). Multi-concept customization of text-to-image diffusion. In Proceedings of the IEEE/CVF Conference on Computer Vision and Pattern Recognition (pp. 1931-1941).
>
> [2] Gal, R., Alaluf, Y., Atzmon, Y., Patashnik, O., Bermano, A. H., Chechik, G., & Cohen-Or, D. (2022). An image is worth one word: Personalizing text-to-image generation using textual inversion. arXiv preprint arXiv:2208.01618.

---

> > ### Author Response · Authors · 2023-11-21
> > **Continue Rebuttal from Authors of Paper4944 to Reviewer 73Bx (2/3)**
> >
> > > **Q3**: The improvement is marginal for both VL and NLP tasks. From Table 1, we can find that MOLE cannot achieve the best text-alignment score for 2/4 cases and cannot achieve the best image-alignment score in the second row for 3/4 cases.
> >
> > > **Q4**: The samples are not enough to support the claims in VL tasks.
> >
> > **A3 & A4 Continue**. Given the absence of suitable LoRA fusion methods for comparison in the Visual & Language (V&L) domain, we introduced two state-of-the-art algorithms for multi-concept fusion that are not LoRA-based: **Custom** $^{[1]}$ and **Textual Inversion** $^{[2]}$. Both of these algorithms focus on fine-tuning all parameters to achieve better results.
> >
> > As illustrated in following Table, our model significantly outperforms existing state-of-the-art LoRA fusion methods. For instance, in terms of text alignment, our MoLE surpasses SVDiff by a margin of 0.031, and in image alignment, it outperforms by 0.028. **Moreover, compared with full-tuning methods (Custom and Textual Inversion), our MoLE also outperforms Textual Inversion in both image alignment and text alignment, and surpasses Custom in text alignment.** This further underscores the effectiveness of our MoLE in comparison to methods involving full parameter tuning.
> >
> > **Text-alignment Results**
> >
> > | Number of concepts | SF      | Custom    | Textual Inversion | SVDiff | Our       |
> > | ------------------ | ------- | --------- | ----------------- | ------ | --------- |
> > | 3                  | 0.678   | 0.751     | 0.709             | 0.728  | **0.759** |
> > | 4                  | 0.681   | **0.735** | 0.721             | 0.717  | 0.725     |
> > | 5                  | 0.652   | 0.731     | 0.704             | 0.723  | **0.762** |
> > | 6                  | 0.678   | 0.722     | **0.735**         | 0.709  | 0.727     |
> > | **Average**        | 0.67225 | 0.734     | 0.717             | 0.719  | **0.752** |
> >
> > **Average Image-alignment Results**
> >
> > | Number of concepts | SF    | Custom    | Textual Inversion | SVDiff       | Our   |
> > | ------------------ | ----- | --------- | ----------------- | ------------ | ----- |
> > | 3                  | 0.694 | **0.761** | 0.720             | 0.719        | 0.757 |
> > | 4                  | 0.712 | **0.760** | 0.736             | 0.721        | 0.742 |
> > | 5                  | 0.682 | **0.798** | 0.710             | 0.708        | 0.737 |
> > | 6                  | 0.698 | 0.721     | **0.747**         | 0.712        | 0.736 |
> > | **Average**        | 0.692 | **0.760** | 0.728             | 0.715 | 0.743 |
> >
> > #### **References**
> >
> >
> > [1] Kumari, N., Zhang, B., Zhang, R., Shechtman, E., & Zhu, J. Y. (2023). Multi-concept customization of text-to-image diffusion. In *Proceedings of the IEEE/CVF Conference on Computer Vision and Pattern Recognition* (pp. 1931-1941).
> >
> > [2] Kumari, N., Zhang, B., Zhang, R., Shechtman, E., & Zhu, J. Y. (2023). Multi-concept customization of text-to-image diffusion. In *Proceedings of the IEEE/CVF Conference on Computer Vision and Pattern Recognition* (pp. 1931-1941).

---

> > > ### Comment · Reviewer_73Bx · 2023-11-22
> > >
> > > Thank you for your insightful responses. However, my concerns regarding the qualitative results persist because no convincing images were generated to support the claim that the mixture of LoRA is effective. The number of samples for quantitative evaluation is still less than 30, which is small. Furthermore, the numeric number is based on the models, which are not straightforward enough to evaluate the quality of the generated images, and the necessity to show some promising generated examples still remains. Hence, I keep my initial scores.

---

### Official Review · Reviewer_Tkgq · 2023-10-31

**Soundness:** 2 fair
**Presentation:** 2 fair
**Contribution:** 2 fair
**Rating:** 5
**Confidence:** 4

**Summary:**

This paper studies the question of how to combine multiple experts adapted from the same pretrained model to maximize generalization of the combination. Particularly, each expert corresponds to adapting the pretrained model using low-rank adaptation (LoRA) on a separate (subset of a) dataset. Unlike prior work that uses a simple addition of multiple LoRA modules or gradient-based methods for combining them, this work proposes a gating mechanism that computes weights for the modules based on the outputs from the pretrained model and each of the modules. Additionally, this work also proposes making the combination more fine-grained, with the mixture weights being layer or block specific in a transformer based model.

Experiments in text-to-image generation and natural language inference (text-only) with 3-4 experts show that the proposed approach outperforms simple combination, and other methods comparable to simple combinations (SVDiff and LoRAHub).

**Strengths:**

- Learning a gating function to combine LoRA modules is a sensible idea and is generally motivated well in the paper.
- The proposed approach does not add too many additional parameters.

**Weaknesses:**

Many details in the paper are unclear
- The related work in Section 2.2 can be more clearly explained. Particularly, it is claimed that the "arithmetic operation-based fusion" suffers from "identity confusion among multiple LoRAs". This issue needs to be clarified. How does the proposed approach fix this issue? Also, the details of the "reference tuning-based fusion" method are unclear. Is the approach from Gu et al., 2023 comparable to this work? If so, why is this approach not compared against them?
- Training method: Section 3.5 says that only the gating parameters were trained, and the others were frozen. Does this mean the LoRA parameters were first individually trained and then kept frozen while the gating parameters were being trained? The current text implies that the LoRA parameters were never trained.
- Experiments: How were the number of experts chosen for both the experiments?
- Results: The results in Table 1 are unclear. What is the difference between LoRA expert 1 vs. expert 2 in the image alignment experiments? Is it the number of experts used? Why are these settings not shown for text alignment?


The experiments, especially the NLI ones, are limited and are not well-motivated.
- Each of the experts is trained on one NLI dataset, and the mixture is evaluated on a mixture of NLI datasets. These are all essentially the same task. What are the experts expected to learn differently in these tasks? It would make more sense to build task specific experts and generalize to new tasks.
- Moreover, the base model used is Flan-T5 which was already trained on all the NLI datasets used in this paper. It would make more sense to adapt the model to other datasets, or use a different base model.

**Questions:**

- In Table 1, results vary across settings (e.g.: robot+dog+clock vs. table+can+teapot) and the baselines are better in some settings. Do you have any insights into why MoLE works better in some settings and not all?
- In Table 2, what is the point of showing results from using different datasets to train the gating parameters? It might be helpful to evaluate how much the performance varies based on the choice of the data used for training the gating function, and check whether the variance is lower for MoLE compared to baselines.
- In Table 2, the labels under the “Model” column in the “In-Domain” setting must be typos. Instead of MoLE^{r}, MoLE^{c} etc., these should be MoLE^{ar1}, MoLE^{ar2} etc. Can you please confirm?
- In both image-generation and NLI experiments, the number of experts trained is between 3 and 5. How do you think the approach would scale to a larger number of experts?

---

> ### Author Response · Authors · 2023-11-21
> **Rebuttal from Authors of Paper4944 to Reviewer Tkgq (1/3)**
>
> Thank you for your detailed, helpful feedback. We address your feedback point by point below.
>
> ------------------------------------------------------------------------------------------------------------------------------------------------------------------------------------
>
> > **Q1**: The related work in Section 2.2 can be more clearly explained. Particularly, it is claimed that the "arithmetic operation-based fusion" suffers from "identity confusion among multiple LoRAs". This issue needs to be clarified. How does the proposed approach fix this issue?
>
>
> The problem of "identity confusion among multiple LoRAs" in arithmetic operation-based fusion is rooted in our
>
> * *Observation 1 (in Section 3.1): "Fusing multiple LoRA candidates directly impacts the generative ability of the original model, while applying weight normalization before the fusion process maintains this generative capacity."*
>
> In the context of the V&L domain, directly merging multiple LoRAs can result in the model producing images with unclear meaning, as depicted in Figure 3 of the main text. Conversely, fusing LoRAs with weight normalization can lead to the model losing the distinctive characteristics of individual LoRAs, which is undesirable.
>
> In the NLP domain, we trained a LoRA on a combination of ANLI-RA, ANLI-R2, and QNLI datasets and observed that using five or more LoRAs caused the model's output to become chaotic, repeating words like 'like, like, like...'. This demonstrates that directly fusing multiple LoRAs in NLP can negatively impact generative abilities. Additionally, when we applied weight normalization to combine LoRAs trained on five datasets (ANLI-R1, ANLI-R2, ANLI-R3, QNLI, and WNLI), we noticed a decrease in performance on four of the datasets, as shown in the table below:
>
> | Model         | ANLI-R1       | ANLI-R2       | ANLI-R3 | QNLI          | WNLI          | Avg           |
> | ------------- | ------------- | ------------- | ------- | ------------- | ------------- | ------------- |
> | Single LoRA   | 80.32         | 79.02         | 75.92   | 78.62         | 74.32         | 77.64         |
> | Simple Fusion | 79.32 (**↓**) | 78.88 (**↓**) | 76.42   | 78.06 (**↓**) | 69.98 (**↓**) | 76.53 (**↓**) |
>
> ***How does the proposed approach fix this issue?***
>
> Our proposed approach is primarily based on
>
> * *Observation 2 (Section 3.1), where we found that different layers of LoRA represent different characteristics.*
>
> In the V&L domain, as illustrated in Figure 2 of the main text, different layers generate different features for target objects. In the NLP domain, as supported by our experiments, different layers of LoRA acquire distinct feature representations, as shown in the table below:
>
> |           | ANLI-R1   | ANLI-R2   | QNLI      |
> | --------- | --------- | --------- | --------- |
> | Full LoRA | 81.65     | 80.03     | 76.42     |
> | 0%-20%    | 78.72     | 78.35     | **78.14** |
> | 20%-40%   | 76.10     | 77.96     | 77.85     |
> | 40%-60%   | 76.95     | **81.47** | 74.57     |
> | 60%-80%   | 77.25     | 78.19     | 75.71     |
> | 80%-100%  | **82.59** | 77.91     | 75.48     |
>
> Our approach leverages this observation to achieve more dynamic fusion. We allocate higher weights to the characteristics needed within each layer and reduce weights for unnecessary features, effectively addressing the issue of identity confusion among multiple LoRAs. This allows us to achieve more tailored and effective fusion within limited parameter capacity.
>
> For instance, in the V&L domain, as depicted in Figure 9 of the main text, our MoLE effectively preserves the characteristics of individual objects. In the NLP domain, as shown in Table 2 of the main text, our MoLE demonstrates comprehensive performance improvements compared to simple direct fusion. These experiments collectively demonstrate that our MoLE can dynamically select and weight LoRA features through hierarchical control, resulting in more effective and meaningful fusion.
>
>
> ------------------------------------------------------------------------------------------------------------------------------------------------------------------------------------
>
> > **Q2**: Also, the details of the "reference tuning-based fusion" method are unclear. Is the approach from Gu et al., 2023 comparable to this work? If so, why is this approach not compared against them?
>
> **A2.** The reference tuning-based fusion method predominantly centers on the V&L domain and necessitates the incorporation of supplementary information, such as deliberately designed object position masks. This methodology introduces a kind of prior knowledge that our model does not depend on. Consequently, conducting a direct comparison between our model and this method may not be entirely equitable, given the disparate assumptions and information inputs involved.

---

> ### Author Response · Authors · 2023-11-21
> **Rebuttal from Authors of Paper4944 to Reviewer Tkgq (2/3)**
>
> > **Q3**: Training method: Section 3.5 says that only the gating parameters were trained, and the others were frozen. Does this mean the LoRA parameters were first individually trained and then kept frozen while the gating parameters were being trained? The current text implies that the LoRA parameters were never trained.
>
> **A3.** Your understanding is correct. The purpose of our paper is to explore a better way to merge LoRA candidates. Here, "LoRA candidates" refer to LoRA models that have been independently trained on different datasets for distinct purposes. After obtaining these LoRA candidates, we then freeze the parameters of them and train the gating function to merge them.
>
>
>
> ------------------------------------------------------------------------------------------------------------------------------------------------------------------------------------
>
> > **Q4**: Results: The results in Table 1 are unclear. What is the difference between LoRA expert 1 vs. expert 2 in the image alignment experiments? Is it the number of experts used? Why are these settings not shown for text alignment?
>
> **A4.** We apologize for any ambiguity in our explanation. In the rows pertaining to image alignment, the expert indices (e.g., expert 1, expert 2) indicate which expert's training images were utilized to calculate the visual similarity with the final generated image. For example, expert 1 might correspond to 'Dog' while expert 2 corresponds to 'Wolf'. The 'text-alignment score' signifies the similarity between the final generated image and the text prompt used to create it.
>
> To clarify further, we employ a specific prompt (e.g., **t**: "a photo of a [𝑉1] dog wearing [𝑉3] sunglasses, with a [𝑉2] cat beside") to generate the final image **F**, which includes elements such as [𝑉1] dog, [𝑉2] cat, and [𝑉3] sunglasses. We determine the similarity between **F** and **t** using CLIP, resulting in the text-alignment score. Additionally, we compute the similarity between **F** and the images corresponding to [𝑉1] dog, [𝑉2] cat, and [𝑉3] sunglasses (corresponding to LoRA expert 1, 2, and 3 as mentioned in the table) to calculate the image-alignment score.
>
> We acknowledge the need for clarification, and we will ensure that this ambiguity is addressed comprehensively in the final version of our paper.
>
> ------------------------------------------------------------------------------------------------------------------------------------------------------------------------------------
>
> > **Q5**: In Table 2, the labels under the “Model” column in the “In-Domain” setting must be typos. Instead of MoLE^{r}, MoLE^{c} etc., these should be MoLE^{ar1}, MoLE^{ar2} etc. Can you please confirm?
>
> **A5.** Thank you for bringing this to our attention, and we appreciate your diligence in reviewing our work. We sincerely apologize for the typos in Table 2. In the 'In-Domain' setting, the labels under the 'Model' column should indeed be MoLE$^{ar1}$, MoLE$^{ar2}$, MoLE$^{ar3}$, and MoLE$^{q}$, respectively. We acknowledge this oversight, and rest assured, we will rectify these errors in the final version of our paper.

---

> ### Author Response · Authors · 2023-11-21
> **Rebuttal from Authors of Paper4944 to Reviewer Tkgq (2.5 /3)**
>
> ------------
>
> > **Q6**: Experiments: How were the number of experts chosen for both the experiments?
>
> **A6.**
>
> * For V&L, we follow the common diffusion multi-concept composition setting$^{[1][2]}$, which usually compose 2~3 concepts into a new multi-concepts image.
> * For NLP domain, We determined the number of LoRA candidates based on the dataset sizes of various subdomains within the FLAN dataset. Specifically, within the NLI-related datasets in FLAN, there are several sub-datasets, including ANLI (R1-R3), RTE, CB, WNLI, QNLI, MNLI, among others. Therefore, we selected a subset (5~6) of these as LoRA candidates, while the remainder served as the training set. Given the limited number of LoRA concepts, we conduct new experiments by extending the number of experts from 5 to 128, the following  results are shown in **A7**.
>
> > **Q7**. In both image-generation and NLI experiments, the number of experts trained is between 3 and 5. How do you think the approach would scale to a larger number of experts?
>
> **A7.**
>
> **NLP domain**
>
> We conducted experiments with an extended number of LoRA (8, 24, 48, 128) in the NLP domain, and the experimental results are as follows Table.
>
> * Our MoLE achieved optimal performance in multiple settings with different numbers of LoRA. Particularly, when the number of LoRA is large, such as 48 and 128, our MoLE outperformed LoRAHub by **2.5** and **3.0**, respectively. Through analysis, we found that LoRAHub's optimization algorithm tends to assign most LoRA weights to zero when there are many LoRA candidates. This limitation hinders LoRAHub from fully utilizing the characteristics of all LoRA. In contrast, our MoLE mitigates this issue.
>
> * Besides, as the number of LoRA increases to a very large value (128), the performance of all methods deteriorates. This is an intriguing phenomenon, indicating that current methods, including our MoLE, do not perform well when dealing with a large number of LoRA. This is an area we plan to explore and address in the future.
>
> | Number of LoRA | Simple Fusion | LoRAHub | Our      |
> | -------------- | ------------- | ------- | -------- |
> | 8              | 30.7          | 31.9    | **34.6** |
> | 24             | 34.8          | 35.1    | **36.7** |
> | 48             | 32.4          | 34.9    | **37.4** |
> | 128            | 32.1          | 33.5    | **36.5** |
>
> *The result is the average EM on the Big-Bench Hard (BBH) dataset*
>
> **V&L domain**
>
> As illustrated in the table below, we systematically increased the number of fused LoRAs in V&L experiments. Typically, experiments that involve fusing multiple visual concepts set the fusion number to be in the range of 3 to 4. Fusing too many concepts can potentially lead to ambiguous model outputs; hence, we set the fusion number in the range of 3 to 6. Our findings consistently demonstrate that as the number of LoRAs increased, our MoLE consistently outperformed other models in terms of average text alignment and image alignment. This observation highlights the robustness of our MoLE to varying numbers of LoRAs and its enhanced fusion capability.
>
>
> | Number of LoRA | Text-alignment (Simple Fusion) | Text-alignment (SVDiff) | Text-alignment (MoLE) | Average Image-alignment (Simple Fusion) | Average Image-alignment (SVDiff) | Average Image-alignment (MoLE) |
> |-----------------|----------------------|-------------------------|---------------------|-----------------------------|--------------------------------|----------------------------|
> | 3               | 0.678                | 0.728                   | **0.759**           | 0.694                       | 0.719                          | **0.757**                  |
> | 4               | 0.681                | 0.717                   | **0.725**           | 0.712                       | 0.721                          | **0.742**                  |
> | 5               | 0.652                | 0.723                   | **0.762**           | 0.682                       | 0.708                          | **0.737**                  |
> | 6               | 0.698                | 0.709                   | **0.737**           | 0.703                       | 0.701                          | **0.709**                  |
> | Average         | 0.677                | 0.719                   | **0.746**           | 0.698                       | 0.712                          | **0.736**                  |
>
>
>
>
> **References**
>
> [1] Kumari, N., Zhang, B., Zhang, R., Shechtman, E., & Zhu, J. Y. (2023). Multi-concept customization of text-to-image diffusion. In *Proceedings of the IEEE/CVF Conference on Computer Vision and Pattern Recognition* (pp. 1931-1941).
>
> [2] Gal, R., Alaluf, Y., Atzmon, Y., Patashnik, O., Bermano, A. H., Chechik, G., & Cohen-Or, D. (2022). An image is worth one word: Personalizing text-to-image generation using textual inversion. arXiv preprint arXiv:2208.01618.

---

> ### Author Response · Authors · 2023-11-21
> **Rebuttal from Authors of Paper4944 to Reviewer Tkgq (2.75 /3)**
>
> -----------------------
> > **Q8**: In Table 1, results vary across settings (e.g.: robot+dog+clock vs. table+can+teapot) and the baselines are better in some settings. Do you have any insights into why MoLE works better in some settings and not all?
>
> **A8**. As shown in following Table, there is usually a trade-off between text-alignment and image-alignment$^{[1][2]}$ on the multi-concepts fusion in V & L domain (The visualization of this trade-off  can be seen in the Figure.12 in the supp). High image-alignment leads to a decrease in text-alignment where sample generations have less variance and are close to input target images. As a result, our MoLE may not perform better in both image and text sub alignment.
>
> | Training Steps | Text Alignment Similarity | Image Alignment Similarity |
> | -------------- | ------------------------- | -------------------------- |
> | 500            | 0.83                      | 0.68                       |
> | 1000           | 0.81                      | 0.73                       |
> | 1500           | 0.80                      | 0.75                       |
> | 2000           | 0.795                     | 0.76                       |
> | 2500           | 0.792                     | 0.77                       |
> | 3000           | 0.786                     | 0.774                      |
>
> However, as depicted in Table 1 in the main text, it is worth noting that when evaluating the overall average across all configurations, our MoLE consistently outperforms both in average text alignment and image alignment. This underscores MoLE's capability to achieve superior alignment between text and images. Furthermore, we present additional results in the V&L domain following to further validate MoLE's effectiveness. In these results, it's evident that MoLE achieves the highest average scores in both image alignment and text alignment.
>
> -----------
>
> | Number of LoRA | Text-alignment (Simple Fusion) | Text-alignment (SVDiff) | Text-alignment (MoLE) | Image-alignment (LoRA expert 1, Simple Fusion) | Image-alignment (LoRA expert 1, SVDiff) | Image-alignment (LoRA expert 1, MoLE) | Image-alignment (LoRA expert 2, Simple Fusion) | Image-alignment (LoRA expert 2, SVDiff) | Image-alignment (LoRA expert 2, MoLE) | Image-alignment (LoRA expert 3, Simple Fusion) | Image-alignment (LoRA expert 3, SVDiff) | Image-alignment (LoRA expert 3, MoLE) |
> |-----------------|-----------------------------|-------------------------|---------------------|--------------------------------------------|------------------------------------------|-------------------------------------|--------------------------------------------|------------------------------------------|-------------------------------------|--------------------------------------------|------------------------------------------|-------------------------------------|
> | Fancy boot + Monster + Clock | 0.754 | 0.742 | 0.832 | 0.781 | 0.758 | 0.784 | 0.791 | 0.749 | 0.801 | 0.763 | 0.812 | 0.809 |
> | Emoji + Car + Cartoon | 0.610 | 0.607 | 0.696 | 0.619 | 0.734 | 0.839 | 0.711 | 0.702 | 0.709 | 0.652 | 0.686 | 0.679 |
> | Vase + Wolf plushie + Teapot | 0.752 | 0.812 | 0.863 | 0.687 | 0.807 | 0.835 | 0.705 | 0.782 | 0.746 | 0.653 | 0.694 | 0.721 |
> | White Cat + Wolf plushie + Can | 0.704 | 0.772 | 0.780 | 0.801 | 0.804 | 0.802 | 0.678 | 0.763 | 0.825 | 0.650 | 0.729 | 0.714 |
> | Shiny sneaker + Wolf plushie + Teapot | 0.778 | 0.789 | 0.791 | 0.812 | 0.783 | 0.690 | 0.723 | 0.751 | 0.790 | 0.688 | 0.676 | 0.721 |
> | Car + Wolf plushie + Teapot | 0.635 | 0.681 | 0.684 | 0.652 | 0.763 | 0.713 | 0.601 | 0.664 | 0.745 | 0.685 | 0.612 | 0.707 |
> | Can + Wolf plushie + backpack | 0.601 | 0.782 | 0.754 | 0.653 | 0.705 | 0.767 | 0.602 | 0.755 | 0.782 | 0.681 | 0.738 | 0.723 |
> | Golden Retriever + Wolf plushie + Teapot | 0.670 | 0.716 | 0.784 | 0.713 | 0.784 | 0.790 | 0.601 | 0.802 | 0.809 | 0.678 | 0.761 | 0.748 |
> | Golden Retriever + Boot + Monster | 0.614 | 0.762 | 0.755 | 0.665 | 0.662 | 0.620 | 0.748 | 0.832 | 0.862 | 0.723 | 0.719 | 0.735 |
> | Backpack dog + Bowl + Teapot | 0.607 | 0.712 | 0.703 | 0.653 | 0.672 | 0.756 | 0.734 | 0.720 | 0.755 | 0.692 | 0.688 | 0.701 |
> | Backpack dog + White Cat + Emoji | 0.648 | 0.703 | 0.717 | 0.674 | 0.692 | 0.812 | 0.719 | 0.741 | 0.701 | 0.742 | 0.720 | 0.796 |
> | Average | 0.678 | 0.728 | **0.759** | 0.715 | 0.746 | **0.783** | 0.682 | 0.731 | **0.756** | 0.686 | 0.708 | **0.732** |
>
>
> **References**
>
> [1] Kumari, N., Zhang, B., Zhang, R., Shechtman, E., & Zhu, J. Y. (2023). Multi-concept customization of text-to-image diffusion. In Proceedings of the IEEE/CVF Conference on Computer Vision and Pattern Recognition (pp. 1931-1941).
>
> [2] Gal, R., Alaluf, Y., Atzmon, Y., Patashnik, O., Bermano, A. H., Chechik, G., & Cohen-Or, D. (2022). An image is worth one word: Personalizing text-to-image generation using textual inversion. arXiv preprint arXiv:2208.01618.

---

> ### Author Response · Authors · 2023-11-21
> **Rebuttal from Authors of Paper4944 to Reviewer Tkgq (2.9 /3)**
>
> > **Q9**: It would make more sense to adapt the model to other datasets, or use a different base model.?
>
> **A9**. We conducted extensive experiments across various tasks, including **Translation**, **Struct to Text**, **Closed-Book QA**, and multiple subtasks within the challenging **Big-Bench Hard (BBH)** datasets. Additionally, for the sake of thorough comparison, we introduced a new LoRA merging variant called **PEMs** $^{[1]}$, recently proposed in the field of Natural Language Processing.
>
> The corresponding experimental results are summarized in the following table. In summary, our MoLE surpasses state-of-the-art LoRA merging variants on four distinct datasets and tasks, showcasing robust performance. With a notable highlight on the BBH dataset, our MoLE achieves an average performance improvement of **3.8** over LoRAHub and outperforms PEMs by a substantial margin of **9.0**.
>
> In the domain of natural language generation tasks (**Translation** and **Struct to Text**), our MoLE consistently demonstrates superior average performance in the Translation task set, surpassing LoRAHub by **1.5** and PEMs by **2.7**. Similarly, within the Struct to Text task set, our model achieves an average performance advantage of **2.1** over LoRAHub and **2.6** over PEMs. These results highlight the effectiveness of our model in generation task.
>
> --------------
>
> | Task                            | Metric  | LoRAHub  | PEMs     | Our      |
> | ------------------------------- | ------- | -------- | -------- | -------- |
> | **Translation**                 |         |          |          |          |
> | WMT '14 En->Fr                  | BLEU    | 27.4     | 25.6     | **29.1** |
> | WMT '14 Fr->En                  | BLEU    | 29.4     | 27.1     | **31.3** |
> | WMT '16 En->De                  | BLEU    | 24.6     | 24.9     | **27.7** |
> | WMT '16 De->En                  | BLEU    | **29.9** | 28       | 29.1     |
> | WMT '16 En->Ro                  | BLEU    | 17.7     | 15.2     | **18.9** |
> | WMT '16 Ro->En                  | BLEU    | 23.5     | 21.7     | **25.1** |
> | Average                         | BLEU    | 25.4     | 24.2     | **26.9** |
> | **Struct to Text**              |         |          |          |          |
> | CommonGen                       | Rouge-1 | 53.7     | 48.8     | **55.1** |
> |                                 | Rouge-2 | **23.1** | 22.4     | 23.1     |
> |                                 | Rouge-L | 49.7     | 47.2     | **53.9** |
> | DART                            | Rouge-1 | 45.3     | 46.2     | **48.8** |
> |                                 | Rouge-2 | 22.6     | 18.9     | **23.5** |
> |                                 | Rouge-L | 35.1     | **37.6** | 36.0     |
> | E2ENLG                          | Rouge-1 | 41.1     | 40.7     | **42.0** |
> |                                 | Rouge-2 | 26.3     | 24.2     | **29.0** |
> |                                 | Rouge-L | 38.8     | **42.1** | 41.8     |
> | WebNLG                          | Rouge-1 | 52.1     | 52.0     | **54.5** |
> |                                 | Rouge-2 | 23.9     | 24.6     | **26.8** |
> |                                 | Rouge-L | 45.2     | **47.8** | 49.3     |
> | Average                         |         | 38.1     | 37.7     | **40.3** |
> | **Closed-Book QA**              |         |          |          |          |
> | ARC-c                           | EM      | 51.7     | 50.4     | **52.9** |
> | ARC-e                           | EM      | 69.7     | 65.7     | **70.3** |
> | NQ                              | EM      | 17.3     | 16.1     | **23.5** |
> | TQA                             | EM      | **54.5** | 53.9     | 54.0     |
> | Average                         | EM      | 48.3     | 46.5     | **50.2** |
> | **Big-Bench Hard (BBH)**        |         |          |          |          |
> | Boolean Expressions             | EM      | 55.1     | 53.0     | **57.3** |
> | Causal Judgement                | EM      | 57.6     | 51.1     | **57.9** |
> | Date Understanding              | EM      | **31.0** | 29.3     | 30.7     |
> | Disambiguation                  | EM      | 46.6     | 47.2     | **49.3** |
> | Penguins in a Table             | EM      | 41.4     | 39.8     | **45.0** |
> | Reasoning about Colored Objects | EM      | 35.2     | **37.5** | 33.7     |
> | Ruin Names                      | EM      | 19.9     | 19.3     | **21.2** |
> | Average                         | EM      | 38.4     | 33.2     | **42.2** |
>
> **References**
>
> [1] Zhang, J., Chen, S., Liu, J., & He, J. (2023). Composing parameter-efficient modules with arithmetic operations. arXiv preprint arXiv:2306.14870.

---

> ### Author Response · Authors · 2023-11-21
> **Rebuttal from Authors of Paper4944 to Reviewer Tkgq (3/3)**
>
> Thank you for your detailed, helpful feedback.
>
> -------
> > **Q10**. Each of the experts is trained on one NLI dataset, and the mixture is evaluated on a mixture of NLI datasets. These are all essentially the same task. What are the experts expected to learn differently in these tasks? It would make more sense to build task specific experts and generalize to new tasks.
>
> > **Q11**. Moreover, the base model used is Flan-T5 which was already trained on all the NLI datasets used in this paper. It would make more sense to adapt the model to other datasets, or use a different base model.
>
> **A10 & A11**. To further validate the effectiveness of our MoLE, we conducted generalization experiments. Specifically, all LoRA candidates and LoRA merging variants, including MoLE and LoRAHub, were trained on NLI tasks such as ANLI-R1, ANLI-R2, ANLI-R3, QNLI, and WNLI, among others. Subsequently, we evaluated these methods on the challenging Big-Bench Hard (BBH) dataset.
>
> As illustrated in following Table, our MoLE consistently outperforms other LoRA merging variants across nearly all sub-tasks. It achieves an average performance advantage of **2.4** over LoRAHub and **3.7** over PEMs, underscoring its robust and superior performance.
>
> -----
> | Task                               | Metric | LoRAHub | PEMs  | **MoLE** |
> |------------------------------------|--------|---------|-------|---------|
> | **Big-Bench Hard~(BBH)**           |        |         |       |         |
> | **Boolean Expressions**            | EM     | 45.3    | 45.5  | **48.7**|
> | **Causal Judgement**               | EM     | **51.3**| 46.1  | **52.4**|
> | **Date Understanding**             | EM     | **27.5**| 24.6  | 26.6    |
> | **Disambiguation**                 | EM     | 39.7    | 42.4  | **43.8**|
> | **Penguins in a Table**           | EM     | 35.3    | 33.6  | **39.0**|
> | **Reasoning about Colored Objects**| EM     | 32.2    | 31.4  | **34.7**|
> | **Average**                        | EM     | 38.5    | 37.2  | **40.9**|
>
> ----
>
> We want to express our sincere gratitude for your review. We apologize for the delayed response, as we have been dedicating an extended amount of time to conducting experiments, which has kept you waiting. Please let us know if any of your points were not addressed properly, or if you have any additional questions.

---

### Official Review · Reviewer_VkSX · 2023-10-31

**Soundness:** 3 good
**Presentation:** 3 good
**Contribution:** 3 good
**Rating:** 6
**Confidence:** 4

**Summary:**

The author regards each low-rank adapters as an individual expert, and proposes mixture of expert to combine multiple LoRAs. The proposed MoLE method achieves better LoRA fusion performance compared to direct arithmetic merging. A gating balance loss is proposed to avoid training only few LoRA experts. Empirical results validate the proposed method.

**Strengths:**

+ The motivation of combining MOE with LoRA is sound. Different from the original LoRA, the LoRA weights from both the attention and mlp layers are regarded as one individual LoRA expert.
+ A penalty loss is proposed to tackle the gating imbalance issue, so that more LoRA experts are well-trained.
+ For text-to-image generation task in the V&L domain, the proposed MoLE achieves better average scores. In figure 9, the generated image follows text instructions better.

**Weaknesses:**

+ Overall, for NLI tasks in NLP domain, the proposed MoLE shows similar average performance compared with LoRAhub.
+ Combining MOE with LoRA seems straightforward.
+ No other LoRA merging variants are compared in experiments.

**Questions:**

+ For the gating imbalance, instead of introducing a new loss, can we just increase the temperature? What's the difference in empirical performance?
+ From the motivation of the proposed MoLE for hierarchical control, do you think it's better suited for the text-to-image generation task than the NLI task?

---

> ### Author Response · Authors · 2023-11-21
> **Rebuttal from Authors of Paper4944 to Reviewer VkSX (1/2)**
>
> We are grateful to the Reviewer for the extensive review. We address your questions point by point below
>
> ------------------------------------------------------------------------------------------------------------------------------------------------------------------------------------
> > **Q1**: From the motivation of the proposed MoLE for hierarchical control, do you think it's better suited for the text-to-image generation task than the NLI task?
>
> **A1.** Our motivation was grounded in the consistent patterns we observed in both the NLP and VL domains. We apologize for the lack of experimental support for our motivations in the main text concerning the NLP domain. Our NLP domain support experiments are as follows:
>
> * *Observation1: Fusing multiple LoRA candidates directly impacts the generative ability of the original model, while applying weight normalization before the fusion process maintains this generative capacity.*
>
> We combined multiple LoRAs with the FLAN-T5 model. When using five or more LoRAs, the model's output became chaotic, repeating words like 'like like like...'. This shows that directly fusing multiple LoRAs in NLP can affect generative abilities. Additionally, we applied weight normalization to combine LoRAs trained on five datasets (ANLI-R1, ANLI-R2, ANLI-R3, QNLI, and WNLI). As shown in the following table, we observed a decrease in performance for the fused LoRA on four of the datasets. This indicates that while weight normalization preserves generative capacity, it also impacts the intrinsic properties of LoRA.
>
>
> | Model         | ANLI-R1       | ANLI-R2       | ANLI-R3 | QNLI          | WNLI          | Avg           |
> | ------------- | ------------- | ------------- | ------- | ------------- | ------------- | ------------- |
> | Single LoRA   | 80.32         | 79.02         | 75.92   | 78.62         | 74.32         | 77.64         |
> | Simple Fusion | 79.32 (**↓**) | 78.88 (**↓**) | 76.42   | 78.06 (**↓**) | 69.98 (**↓**) | 76.53 (**↓**) |
>
>
> * *Observation2: Different layers of LoRA learn distinct feature representations, collectively forming the overall identity of LoRA.*
>
> We trained a LoRA on a combination of ANLI-RA, ANLI-R2, and QNLI datasets, as shown in the table below (Visualized results can be found in Figure 11 of the PDF). We observed significant performance variations across different layers of this LoRA when evaluated on these sub-datasets. Specifically, the 0%\~20% layers performed best on QNLI, the 40%\~60% layers excelled on ANLI-R2, and the 80%\~100% layers outperformed others on ANLI-R1. This reinforces the idea that in the NLP domain, different layers of LoRA acquire distinct feature representations.
>
> |           | ANLI-R1   | ANLI-R2   | QNLI      |
> | --------- | --------- | --------- | --------- |
> | Full LoRA | 81.65     | 80.03     | 76.42     |
> | 0%-20%    | 78.72     | 78.35     | **78.14** |
> | 20%-40%   | 76.10     | 77.96     | 77.85     |
> | 40%-60%   | 76.95     | **81.47** | 74.57     |
> | 60%-80%   | 77.25     | 78.19     | 75.71     |
> | 80%-100%  | **82.59** | 77.91     | 75.48     |
>
> Based on the two sets of experiments presented above, we can conclude that the motivations we proposed in Section 3.1 hold true in the NLP domain as well. We will include the supporting experiments in the NLP domain mentioned above in the final version to substantiate our observations
>
>
>
> ------------------------------------------------------------------------------------------------------------------------------------------------------------------------------------
>
> > **Q2**: Combining MoE with LoRA seems straightforward.
>
> **A2.**  We  thank the reviewer for valuable insights. The main motivation behind our work is not simply to integrate Mixture-of-Experts (MoE) with LoRA, but rather to build upon our two observations (Section 3.1).
>
> * *Observation1: Fusing multiple LoRA candidates directly impacts the generative ability of the original model, while applying weight normalization before the fusion process maintains this generative capacity.*
>
> * *Observation2: Different layers of LoRA learn distinct feature representations, collectively forming the overall identity of LoRA.*
>
> Firstly, we recognize that LoRA represents distinct features across different layers. Leveraging this insight, we believe that introducing hierarchical control can enable more flexible and dynamic combinations of LoRA while preserving their individual characteristics—a common challenge faced by the research community. In this context, the MoE serves as a valuable tool to facilitate this hierarchical control, effectively harnessing this property to address the challenging problem of LoRA fusion. **In summary, our primary contribution lies in pioneering the exploration of dissecting and layering LoRA for the first time and utilizing this observation to achieve more dynamic LoRA combinations. MoE is a tool we employ to realize this vision**.

---

> ### Author Response · Authors · 2023-11-21
> **Rebuttal from Authors of Paper4944 to Reviewer VkSX (1.5/2)**
>
> > **Q3**: Overall, for NLI tasks in NLP domain, the proposed MoLE shows similar average performance compared with LoRAhub.
> >  **Q4**: No other LoRA merging variants are compared in experiments.
>
> In the NLP domain, we conducted comparisons between our MoLE model and the state-of-the-art LoRA merging variant, **LoRAHub** $^{[1]}$ (submitted to ICLR24, too). In the V&L domain, we compared our MoLE with the leading LoRA merging variant, **SVDiff** $^{[2]}$. Our results demonstrate that in both domains, our MoLE consistently achieves superior performance.
> To further validate the effectiveness of our MoLE, we have conducted additional experiments, including comparisons with recently-proposed LoRA merging variant baselines.
>
> * **NLP domain**
>
> We conducted extensive experiments across various tasks, including **Translation**, **Struct to Text**, **Closed-Book QA**, and multiple subtasks within the challenging **Big-Bench Hard (BBH)** datasets. Additionally, we introduced a new LoRA merging variant called **PEMs** $^{[3]}$, recently proposed in the field of NLP.
>
> The experimental results are summarized in the following table. In summary, our MoLE surpasses state-of-the-art LoRA merging variants on four distinct datasets and tasks, showcasing robust performance. With a notable highlight on the BBH dataset, our MoLE achieves an average performance improvement of **3.8** over LoRAHub and outperforms PEMs by a substantial margin of **9.0**.
>
> In the domain of natural language generation tasks (**Translation** and **Struct to Text**), our MoLE consistently demonstrates superior average performance in the Translation task set, surpassing LoRAHub by **1.5** and PEMs by **2.7**. Similarly, within the Struct to Text task set, our model achieves an average performance advantage of **2.1** over LoRAHub and **2.6** over PEMs. These results highlight the effectiveness of our model in generation task.
>
> | Task                            | Metric  | LoRAHub  | PEMs     | Our      |
> | ------------------------------- | ------- | -------- | -------- | -------- |
> | **Translation**                 |         |          |          |          |
> | WMT '14 En->Fr                  | BLEU    | 27.4     | 25.6     | **29.1** |
> | WMT '14 Fr->En                  | BLEU    | 29.4     | 27.1     | **31.3** |
> | WMT '16 En->De                  | BLEU    | 24.6     | 24.9     | **27.7** |
> | WMT '16 De->En                  | BLEU    | **29.9** | 28       | 29.1     |
> | WMT '16 En->Ro                  | BLEU    | 17.7     | 15.2     | **18.9** |
> | WMT '16 Ro->En                  | BLEU    | 23.5     | 21.7     | **25.1** |
> | Average                         | BLEU    | 25.4     | 24.2     | **26.9** |
> | **Struct to Text**              |         |          |          |          |
> | CommonGen                       | Rouge-1 | 53.7     | 48.8     | **55.1** |
> |                                 | Rouge-2 | **23.1** | 22.4     | 23.1     |
> |                                 | Rouge-L | 49.7     | 47.2     | **53.9** |
> | DART                            | Rouge-1 | 45.3     | 46.2     | **48.8** |
> |                                 | Rouge-2 | 22.6     | 18.9     | **23.5** |
> |                                 | Rouge-L | 35.1     | **37.6** | 36.0     |
> | E2ENLG                          | Rouge-1 | 41.1     | 40.7     | **42.0** |
> |                                 | Rouge-2 | 26.3     | 24.2     | **29.0** |
> |                                 | Rouge-L | 38.8     | **42.1** | 41.8     |
> | WebNLG                          | Rouge-1 | 52.1     | 52.0     | **54.5** |
> |                                 | Rouge-2 | 23.9     | 24.6     | **26.8** |
> |                                 | Rouge-L | 45.2     | **47.8** | 49.3     |
> | Average                         |         | 38.1     | 37.7     | **40.3** |
> | **Closed-Book QA**              |         |          |          |          |
> | ARC-c                           | EM      | 51.7     | 50.4     | **52.9** |
> | ARC-e                           | EM      | 69.7     | 65.7     | **70.3** |
> | NQ                              | EM      | 17.3     | 16.1     | **23.5** |
> | TQA                             | EM      | **54.5** | 53.9     | 54.0     |
> | Average                         | EM      | 48.3     | 46.5     | **50.2** |
> | **Big-Bench Hard (BBH)**        |         |          |          |          |
> | Boolean Expressions             | EM      | 55.1     | 53.0     | **57.3** |
> | Causal Judgement                | EM      | 57.6     | 51.1     | **57.9** |
> | Date Understanding              | EM      | **31.0** | 29.3     | 30.7     |
> | Disambiguation                  | EM      | 46.6     | 47.2     | **49.3** |
> | Penguins in a Table             | EM      | 41.4     | 39.8     | **45.0** |
> | Reasoning about Colored Objects | EM      | 35.2     | **37.5** | 33.7     |
> | Ruin Names                      | EM      | 19.9     | 19.3     | **21.2** |
> | Average                         | EM      | 38.4     | 33.2     | **42.2** |

---

> ### Author Response · Authors · 2023-11-21
> **Continue  Rebuttal from Authors of Paper4944 to Reviewer VkSX (1.5/2)**
>
> **V&L domain**
>
> Given the absence of suitable LoRA fusion methods for comparison in the V&L domain, we introduced two state-of-the-art algorithms for multi-concept fusion that are not LoRA-based: **Custom** $^{[4]}$ and **Textual Inversion** $^{[5]}$. Both of these algorithms focus on fine-tuning all parameters to achieve better results.
>
> As illustrated in following Table, our model significantly outperforms existing state-of-the-art LoRA fusion methods. For instance, in terms of text-alignment, our MoLE surpasses SVDiff by a margin of 0.031, and in image-alignment, it outperforms by 0.028. **Moreover, compared with full-tuning methods (Custom and Textual Inversion), our MoLE also outperforms Textual Inversion in both image-alignment and text-alignment, and surpasses Custom in text-alignment.** This further underscores the effectiveness of our MoLE in comparison to methods involving full parameter tuning.
>
> ------------------------------------------
> **Text-alignment Results**
>
> | Number of concepts | SF      | Custom    | Textual Inversion | SVDiff | Our       |
> | ------------------ | ------- | --------- | ----------------- | ------ | --------- |
> | 3                  | 0.678   | 0.751     | 0.709             | 0.728  | **0.759** |
> | 4                  | 0.681   | **0.735** | 0.721             | 0.717  | 0.725     |
> | 5                  | 0.652   | 0.731     | 0.704             | 0.723  | **0.762** |
> | 6                  | 0.678   | 0.722     | **0.735**         | 0.709  | 0.727     |
> | **Average**        | 0.67225 | 0.734     | 0.717             | 0.719  | **0.752** |
> ----------------------------------------------
>
> **Average Image-alignment Results**
>
> | Number of concepts | SF    | Custom    | Textual Inversion | SVDiff       | Our   |
> | ------------------ | ----- | --------- | ----------------- | ------------ | ----- |
> | 3                  | 0.694 | **0.761** | 0.720             | 0.719        | 0.757 |
> | 4                  | 0.712 | **0.760** | 0.736             | 0.721        | 0.742 |
> | 5                  | 0.682 | **0.798** | 0.710             | 0.708        | 0.737 |
> | 6                  | 0.698 | 0.721     | **0.747**         | 0.712        | 0.736 |
> | **Average**        | 0.692 | **0.760** | 0.728             | 0.715 | 0.743 |
>
> #### **References**
>
> [1] Huang, C., Liu, Q., Lin, B. Y., Pang, T., Du, C., & Lin, M. (2023). Lorahub: Efficient cross-task generalization via dynamic lora composition. *arXiv preprint arXiv:2307.13269*.
>
> [2] Han, L., Li, Y., Zhang, H., Milanfar, P., Metaxas, D., & Yang, F. (2023). Svdiff: Compact parameter space for diffusion fine-tuning. *arXiv preprint arXiv:2303.11305*.
>
> [3] Zhang, J., Chen, S., Liu, J., & He, J. (2023). Composing parameter-efficient modules with arithmetic operations. *arXiv preprint arXiv:2306.14870*.
>
> [4] Kumari, N., Zhang, B., Zhang, R., Shechtman, E., & Zhu, J. Y. (2023). Multi-concept customization of text-to-image diffusion. In *Proceedings of the IEEE/CVF Conference on Computer Vision and Pattern Recognition* (pp. 1931-1941).
>
> [5] Gal, R., Alaluf, Y., Atzmon, Y., Patashnik, O., Bermano, A. H., Chechik, G., & Cohen-Or, D. (2022). An image is worth one word: Personalizing text-to-image generation using textual inversion. arXiv preprint arXiv:2208.01618.

---

> ### Author Response · Authors · 2023-11-22
> **Rebuttal from Authors of Paper4944 to Reviewer VkSX (2/2)**
>
> -------------------
> > **Q4**: For the gating imbalance, instead of introducing a new loss, can we just increase the temperature? What's the difference in empirical performance?
>
> **A4**. Thank you for the inspiration provided. We conducted relevant experiments on the NLI dataset, and after increasing the temperature (denoted as MoLE$^{T1}$, MoLE$^{T2}$, MoLE$^{T3}$ where T1 < T2 < T3), we observed a decrease in the model's performance with temperature increasing, with it becoming more similar to Simple Fusion in terms of performance. We believe that while increasing the temperature can alleviate the imbalance issue in the gating network, it also limits our MoLE from learning more dynamic LoRA combinations, leading to suboptimal results.
>
> | Model         | ANLI-R1       | ANLI-R2       | ANLI-R3 | QNLI          | WNLI          | Avg           |
> | ------------- | ------------- | ------------- | ------- | ------------- | ------------- | ------------- |
> | Simple Fusion | 79.32 | 78.88 | 76.42   | 78.06 | 69.98 | 76.53 |
> |  MoLE   | **81.49** | **79.38** | **77.63** | **79.52** | **72.31** | **78.07** |
> | MoLE$^{T1}$ | 80.52  | 79.27 | 77.30 | 79.11 | 71.07 |  77.45 |
> | MoLE$^{T2}$   | 80.01  | 79.03 | 76.33 |  77.81 |  70.37 |  76.71  |
> | MoLE$^{T3}$   | 78.50   | 79.20   | 76.07   | 78.02 | 70.00 | 76.35 |
>
>
> ----------------
> We want to express our sincere gratitude for your review. We apologize for the delayed response, as we have been dedicating an extended amount of time to conducting experiments, which has kept you waiting. Please let us know if any of your points were not addressed properly, or if you have any additional questions.

---

### Official Review · Reviewer_Qj9C · 2023-11-03

**Soundness:** 3 good
**Presentation:** 2 fair
**Contribution:** 2 fair
**Rating:** 6
**Confidence:** 4

**Summary:**

This paper proposes technique to combine various LoRA (corresponding to different characteristics) via Mixture Of LoRA Experts (MOLE). The technique overcomes the limitations of existing methods of combining LoRAs. The main idea is to combine different LoRAs via a learnable gating mechanism.

**Strengths:**

1. The idea of combining different LoRA via gating mechanism is intuitive and novel.
2. Authors perform extensive set of experiments and show the effectiveness of the technique both in Vision and NLP domain.
3. Authors perform a detailed ablation study to assess various losses and different components.

**Weaknesses:**

1. Authors motivate (section 3.1) the need for Mixture of LoRA for the vision domain but it is not clear if it is also required for the NLP domain as well or not (as also indicated by marginal improvement in results).
2. For the NLP domain the evaluation is done only for one classification task (NLI) and no generative task (e.g., summarization or translation) is evaluated. Analogous to vision domain, it would be great to see effect would MOLE bring in during generation.

**Questions:**

Did authors conduct experiments on the task of machine translation or summarization or any other text generation task?

---

> ### Author Response · Authors · 2023-11-21
> **Rebuttal from Authors of Paper4944 to Reviewer Qj9C (1/2)**
>
> We thank the Reviewer for the useful suggestions. Regarding the questions 1 raised:
>
> > **Q1**: Authors motivate (section 3.1) the need for Mixture of LoRA for the vision domain but it is not clear if it is also required for the NLP domain as well or not (as also indicated by marginal improvement in results).
>
> **A1.** To validate our motivations in the NLP domain, we conducted the following experiments to confirm that our two observations also hold true in the NLP domain:
>
> * *Observation1: Fusing multiple LoRA candidates directly impacts the generative ability of the original model, while applying weight normalization before the fusion process maintains this generative capacity.*
>
> We combined multiple LoRAs with the FLAN-T5 model. When using five or more LoRAs, the model's output became chaotic, repeating words like 'like like like...'. This shows that directly fusing multiple LoRAs in NLP can affect generative abilities. Additionally, we applied weight normalization to combine LoRAs trained on five datasets (ANLI-R1, ANLI-R2, ANLI-R3, QNLI, and WNLI). As shown in the following table, we observed a decrease in performance for the fused LoRA on four of the datasets. This indicates that while weight normalization preserves generative capacity, it also impacts the intrinsic properties of LoRA.
>
>
> | Model         | ANLI-R1       | ANLI-R2       | ANLI-R3 | QNLI          | WNLI          | Avg           |
> | ------------- | ------------- | ------------- | ------- | ------------- | ------------- | ------------- |
> | Single LoRA   | 80.32         | 79.02         | 75.92   | 78.62         | 74.32         | 77.64         |
> | Simple Fusion | 79.32 (**↓**) | 78.88 (**↓**) | 76.42   | 78.06 (**↓**) | 69.98 (**↓**) | 76.53 (**↓**) |
>
>
> * *Observation2: Different layers of LoRA learn distinct feature representations, collectively forming the overall identity of LoRA.*
>
> We trained a LoRA on a combination of ANLI-RA, ANLI-R2, and QNLI datasets, as shown in the table below (Visualized results can be found in Figure 11 of the PDF). We observed significant performance variations across different layers of this LoRA when evaluated on these sub-datasets. Specifically, the 0%\~20% layers performed best on QNLI, the 40%\~60% layers excelled on ANLI-R2, and the 80%\~100% layers outperformed others on ANLI-R1. This reinforces the idea that in the NLP domain, different layers of LoRA acquire distinct feature representations.
>
> |           | ANLI-R1   | ANLI-R2   | QNLI      |
> | --------- | --------- | --------- | --------- |
> | Full LoRA | 81.65     | 80.03     | 76.42     |
> | 0%-20%    | 78.72     | 78.35     | **78.14** |
> | 20%-40%   | 76.10     | 77.96     | 77.85     |
> | 40%-60%   | 76.95     | **81.47** | 74.57     |
> | 60%-80%   | 77.25     | 78.19     | 75.71     |
> | 80%-100%  | **82.59** | 77.91     | 75.48     |
>
> Based on the two sets of experiments presented above, we can conclude that the motivations we proposed in Section 3.1 hold true in the NLP domain as well.

---

> ### Author Response · Authors · 2023-11-21
> **Rebuttal from Authors of Paper4944 to Reviewer Qj9C (2/2)**
>
> We thank the Reviewer for the insightful suggestions. Regarding the questions 2 raised:
>
> -------
> > **Q2**: Did authors conduct experiments on the task of machine translation or summarization or any other text generation task?
>
> **A2**. We conducted extensive experiments across various tasks, including **Translation**, **Struct to Text**, **Closed-Book QA**, and multiple subtasks within the challenging **Big-Bench Hard (BBH)** datasets. Additionally, we introduced a new LoRA merging variant called **PEMs** $^{[1]}$, recently proposed in the field of NLP.
>
> The corresponding experimental results are summarized in the following table. In summary, our MoLE surpasses state-of-the-art LoRA merging variants on four distinct datasets and tasks, showcasing robust performance. With a notable highlight on the BBH dataset, our MoLE achieves an average performance improvement of **3.8** over LoRAHub and outperforms PEMs by a substantial margin of **9.0**.
>
> In the domain of natural language generation tasks (**Translation** and **Struct to Text**), our MoLE consistently demonstrates superior average performance in the Translation task set, surpassing LoRAHub by **1.5** and PEMs by **2.7**. Similarly, within the Struct to Text task set, our model achieves an average performance advantage of **2.1** over LoRAHub and **2.6** over PEMs. These results highlight the effectiveness of our model in generation task.
>
>
>
> | Task                            | Metric  | LoRAHub  | PEMs     | Our      |
> | ------------------------------- | ------- | -------- | -------- | -------- |
> | **Translation**                 |         |          |          |          |
> | WMT '14 En->Fr                  | BLEU    | 27.4     | 25.6     | **29.1** |
> | WMT '14 Fr->En                  | BLEU    | 29.4     | 27.1     | **31.3** |
> | WMT '16 En->De                  | BLEU    | 24.6     | 24.9     | **27.7** |
> | WMT '16 De->En                  | BLEU    | **29.9** | 28       | 29.1     |
> | WMT '16 En->Ro                  | BLEU    | 17.7     | 15.2     | **18.9** |
> | WMT '16 Ro->En                  | BLEU    | 23.5     | 21.7     | **25.1** |
> | Average                         | BLEU    | 25.4     | 24.2     | **26.9** |
> | **Struct to Text**              |         |          |          |          |
> | CommonGen                       | Rouge-1 | 53.7     | 48.8     | **55.1** |
> |                                 | Rouge-2 | **23.1** | 22.4     | 23.1     |
> |                                 | Rouge-L | 49.7     | 47.2     | **53.9** |
> | DART                            | Rouge-1 | 45.3     | 46.2     | **48.8** |
> |                                 | Rouge-2 | 22.6     | 18.9     | **23.5** |
> |                                 | Rouge-L | 35.1     | **37.6** | 36.0     |
> | E2ENLG                          | Rouge-1 | 41.1     | 40.7     | **42.0** |
> |                                 | Rouge-2 | 26.3     | 24.2     | **29.0** |
> |                                 | Rouge-L | 38.8     | **42.1** | 41.8     |
> | WebNLG                          | Rouge-1 | 52.1     | 52.0     | **54.5** |
> |                                 | Rouge-2 | 23.9     | 24.6     | **26.8** |
> |                                 | Rouge-L | 45.2     | **47.8** | 49.3     |
> | Average                         |         | 38.1     | 37.7     | **40.3** |
> | **Closed-Book QA**              |         |          |          |          |
> | ARC-c                           | EM      | 51.7     | 50.4     | **52.9** |
> | ARC-e                           | EM      | 69.7     | 65.7     | **70.3** |
> | NQ                              | EM      | 17.3     | 16.1     | **23.5** |
> | TQA                             | EM      | **54.5** | 53.9     | 54.0     |
> | Average                         | EM      | 48.3     | 46.5     | **50.2** |
> | **Big-Bench Hard (BBH)**        |         |          |          |          |
> | Boolean Expressions             | EM      | 55.1     | 53.0     | **57.3** |
> | Causal Judgement                | EM      | 57.6     | 51.1     | **57.9** |
> | Date Understanding              | EM      | **31.0** | 29.3     | 30.7     |
> | Disambiguation                  | EM      | 46.6     | 47.2     | **49.3** |
> | Penguins in a Table             | EM      | 41.4     | 39.8     | **45.0** |
> | Reasoning about Colored Objects | EM      | 35.2     | **37.5** | 33.7     |
> | Ruin Names                      | EM      | 19.9     | 19.3     | **21.2** |
> | Average                         | EM      | 38.4     | 33.2     | **42.2** |
>
> **References**
>
> [1] Zhang, J., Chen, S., Liu, J., & He, J. (2023). Composing parameter-efficient modules with arithmetic operations. arXiv preprint arXiv:2306.14870.
>
> ------
> We want to express our sincere gratitude for your review. We apologize for the delayed response, as we have been dedicating an extended amount of time to conducting experiments, which has kept you waiting. Please let us know if any of your points were not addressed properly, or if you have any additional questions.

---

> > ### Comment · Reviewer_Qj9C · 2023-11-23
> >
> > Thanks for the detailed response to the queries. The results shown by experiments using MOLE shows the use case for the NLP domain as well. I am increasing my scores.

---

### Public Comment · ~Zeyu_Lu1 · 2023-11-25
**Omitting relevant work in this article**

Hi authors,

After carefully reading your article, I think you missed an article related to this article.
We have already proposed the method of using parameter-efficient weight weight to ensemble in our ICML paper for early **April 2023**. Please refer to it for details. https://proceedings.mlr.press/v202/wu23t/wu23t.pdf

If possible, please cite our article.

Thank you!

---

### Public Comment · ~Arshad1 · 2024-03-19
**Learnable parameter of Gating Function**

Hi authors,
I am curious about replicating your paper. Could you please help me with the following query?

The learnable parameter of the gating function depends on **(N × L × d)**, where **L** represents the sequence length. Since the sequence length **L** can vary during inference, the flattened dimension **ξ** would also be variable.

- How is this handled during inference, considering that during training, we learned the parameter **e** of a fixed size?
- Given that the learnable parameter **e** should have fixed dimensions, is there any masking strategy used to mask out the effect of pad tokens during training, considering that sequences can vary in length?




      where EΩ (x) ∈ RN ×L×d and ⊕ indicates the concatenation operation. Then we flatten and reduce the EΩ (x) to N -dimensions by a dot-product operation with e ∈ Rξ×N , where e is the learnable parameter for the gating function G (·) and ξ = N × L × d,
          ε=Flatten(EΩ(x))⊤ ·e, ε∈RN,     (10)

---

### Meta-Review · Area_Chair_mdcu · 2023-12-08

**Metareview:**

This paper considers the problem of combining many LoRA-based adapters of a base model into a single mixture-of-experts style model. To do so, all model parameters are fixed and a new gating network is trained to weight the outputs of the different (LoRA) experts. The gating module is trained on a target task with a load balancing loss and, in the case of image generation, a domain-specific CLIP-based loss. The resulting approach is tested on image generation models and on some NLP tasks. Originally, reviewers were concerned about the evaluation was limited, with particular issue in the NLP setting where only NLI was considered. In the rebuttal, the authors provided extensive additional experiments, so I think the paper should be accepted as long as all of these experiments make it into the paper.

**Justification For Why Not Higher Score:**

While the paper is convincing, it highlights results on image generation tasks, which are notoriously hard to evaluate. Past works like LoRAHub have done more extensive evaluation. It's also not clear whether practitioners would use this given that it requires training. But, it's an interesting method along an increasingly important line of work.

**Justification For Why Not Lower Score:**

Reviewer consensus was generally to accept.

---

### Decision · Program_Chairs · 2024-01-16

Accept (poster)